# Intestinal NUCB2/nesfatin-1 regulates hepatic glucose production via the MC4R-cAMP-GLP-1 pathway

Shan Geng [1,7], Shan Yang[1,7], Xuejiao Tang[1], Shiyao Xue[1], Ke Li[1], Dongfang Liu[1], Chen Chen [2], Zhiming Zhu[3], Hongting Zheng [4], Yuanqiang Wang[5], Gangyi Yang [1✉], Ling Li [6✉] & Mengliu Yang [1✉]

## Abstract

**Communication of gut hormones with the central nervous system is important to regulate systemic glucose homeostasis, but the precise underlying mechanism involved remain little understood. Nesfatin-1, encoded by nucleobindin-2 (NUCB2), a potent anorexigenic peptide hormone, was found to be released from the gastrointestinal tract, but its specific function in this context remains unclear. Herein, we found that gut nesfatin-1 can sense nutrients such as glucose and lipids and subsequently decreases hepatic glucose production. Nesfatin-1 infusion in the small intestine of NUCB2-knockout rats reduced hepatic glucose production via a gut – brain – liver circuit. Mechanistically, NUCB2/nesfatin-1 interacted directly with melanocortin 4 receptor (MC4R) through its H-F-R domain and increased cyclic adenosine monophosphate (cAMP) levels and glucagon-like peptide 1 (GLP-1) secretion in the intestinal epithelium, thus inhibiting hepatic glucose production. The intestinal nesfatin-1 -MC4R-cAMP-GLP-1 pathway and systemic gut-brain communication are required for nesfatin-1 - mediated regulation of liver energy metabolism. These findings reveal a novel mechanism of hepatic glucose production control by gut hormones through the central nervous system.**

**Keywords** NUCB2/nesfatin-1; Melanocortin 4 Receptor (MC4R); Gut Hormones; Gut-Brain-Liver Neural Circuit; Hepatic Glucose Production (HGP)
**Subject Categories** Metabolism; Neuroscience

## Introduction

Nutrient sensing, an intricately orchestrated physiological mechanism, governs the adaptive responses of the body to available nutrients (Duca et al, 2021), is pertinent to absorption, digestion, and food intake modulation, and is fundamental to maintaining the balance between glucose and energy homeostasis within animal organisms (Chopra et al, 2022). At the interface of nutrient sensing within the gastrointestinal (GI) tract, enteroendocrine cells emerge as pivotal players orchestrating physiological responses (Atanga et al, 2023). These cells synthesize and secrete various hormones, such as glucagon-like peptide 1 (GLP-1), glucose-dependent insulinotropic polypeptide (GIP), ghrelin, and cholecystokinin (CCK) (Atanga et al, 2023), in response to nutrient perturbations, thereby anchoring regulatory networks that preside over appetite, glucose metabolism, and insulin sensitivity. By sensing the nutrient content of the diet, the intestine can provide feedback to the brain, activate the central nervous system (CNS) nutrient-sensing mechanism, and regulate food intake and hepatic glucose production (HGP) (Romani-Perez et al, 2021). The CNS can sense the levels of various antidiabetic drugs (such as metformin and resveratrol), nutrients (such as fatty acids and amino acids), and gut hormones to regulate HGP and lipid metabolism via the afferent and efferent vagal nerves, forming a gut-brain-liver neural circuit (Cote et al, 2015; Duca et al, 2015; Lin et al, 2019; Yang et al, 2017). Dysregulation of nutrient sensing in the intestine can contribute to metabolic disorders, including type 2 diabetes (T2D) and nonalcoholic fatty liver disease (NAFLD) (Duca et al, 2021). Therefore, understanding these mechanisms at the molecular level and the hormones involved can provide invaluable insights into potential therapeutic strategies to combat these diseases.

Nesfatin-1 is an 82 amino acid peptide derived from its precursor, NUCB2 (Shimizu et al, 2009b), and plays a crucial role in various physiological processes. It is widely distributed in the brain and peripheral organs, such as the stomach, small intestine,

[1]Department of Endocrinology, the Second Affiliated Hospital, Chongqing Medical University, Chongqing, China. [2]Endocrinology, SBMS, Faculty of Medicine, University of Queensland, Brisbane, QLD 4072, Australia. [3]Department of Hypertension and Endocrinology, Daping Hospital, Third Military Medical University, Chongqing Institute of Hypertension, Chongqing, China. [4]Department of Endocrinology, Xinqiao Hospital, Third Military Medical University, Chongqing, China. [5]School of Pharmacy and Bioengineering, Chongqing University of Technology, Chongqing 400054, China. [6]The Key Laboratory of Laboratory Medical Diagnostics in the Ministry of Education and Department of Clinical Biochemistry, College of Laboratory Medicine, Chongqing Medical University, Chongqing, China. [7]These authors contributed equally: Shan Geng, Shan Yang.
✉E-mail: gangyiyang@hospital.cqmu.edu.cn; liling@cqmu.edu.cn; mengliu.yang@cqmu.edu.cn

pancreas, adrenal gland, testes, liver, fat, and heart (Prinz et al, 2016; Schalla et al, 2020), indicating that this molecule may have important metabolic, immune, and reproductive functions. In animals, nesfatin-1 treatment has been shown to affect glucose homeostasis, GI function, water consumption, temperature regulation, sleep, and lipid browning (Shimizu et al, 2009b; Wang et al, 2016). One of our preliminary studies revealed elevated nesfatin-1 levels in patients with T2D and impaired glucose tolerance (IGT), which correlated with insulin resistance (IR) (Zhang et al, 2012). In subsequent studies, we emphasized the essential role of central nesfatin-1 signaling in modulating hepatic insulin signaling and glucose metabolism (Wu et al, 2014; Yang et al, 2012), highlighting its importance in the complex network regulating energy balance. Notably, the main sites of NUCB2/nesfatin-1 expression are gastric endocrine X/A-like cells (Stengel et al, 2009), suggesting that the GI tract may be the major source of nesfatin-1 in the circulation. In addition, the localization of NUCB2/nesfatin-1 in the GI tract and its co-localization with gut hormones such as CCK, peptide YY (PYY), and GLP-1 (Ramesh et al, 2015; Ramesh et al, 2016) suggest its potential role in metabolic processes. Thus, the study of intestinal NUCB2/nesfatin-1 could elucidate its role in hepatic glucose regulation and IR, offering new insights into T2D.

The aim of this study was to investigate the molecular and neural mechanisms underlying the regulatory effects of nesfatin-1 on hepatic glucose metabolism and insulin response. Our results suggest that intestinal NUCB2/nesfatin-1 infusion inhibits HGP and enhances insulin sensitivity, a process regulated by melanocortin 4 receptor (MC4R)-cyclic AMP (cAMP)-GLP-1 signaling via gut-brain-liver neural circuitry.

# Results

## The role of NUCB2/nesfatin-1 in modulating intestinal nutrient sensing and the metabolic response

To elucidate the potential role of NUCB2/nesfatin-1 in intestinal nutrient sensing, an initial investigation was undertaken in which glucose or saline was infused into the intestines of wild-type (WT) rats to observe the impact of intestinal nutrients on NUCB2/nesfatin-1 expression across diverse tissues and organs (Fig. 1A). Our findings indicate an increase in the protein expression of NUCB2/nesfatin-1 within the intestine upon glucose infusion. However, comparable alterations were absent in the other tissues or organs (Fig. 1B,C). These preliminary findings suggest that shifts in intestinal nutrient concentrations drive the expression of intestinal NUCB2/nesfatin-1.

Having established the role of NUCB2/nesfatin-1 in nutrient sensing, we expanded our research to explore the metabolic consequences associated with its deficiency. To do this, we employed an NUCB2/nesfatin-1 global knockout (NUCB2-KO) rat model to evaluate alterations in hepatic glucose metabolism in response to duodenal nutrient infusion under conditions of hereditary NUCB2/nesfatin-1 deficiency. As expected, NUCB2-KO rats presented significantly lower NUCB2 mRNA and protein expression across various tissues compared to WT rats (Fig. S1A,B). To perform the evaluation, we implemented a pancreatic-euglycemic clamping (PEC) procedure on WT and NUCB2-KO rats maintained on a normal chow diet (NCD), subjecting them to

duodenal infusion of either glucose or saline (Fig. 1D). Throughout the PEC period, blood glucose, insulin, free fatty acid (FFA), and triglyceride (TG) levels remained stable (Fig. S2A–E). A noteworthy increase in the intravenous glucose infusion rate (GIR) and a decrease in HGP were observed in NCD-fed WT rats with glucose infusion compared to those with saline infusion (Fig. 1E–H). This implies that intestinal glucose sensing inhibits endogenous glucose production in the stable PEC state. In contrast, NUCB2-KO rats demonstrated a substantial reduction in GIR (Fig. 1E,F) and a corresponding impairment in the suppression of HGP (Fig. 1G,H) compared to WT rats during duodenal glucose infusion. Whole-body glucose uptake remained relatively stable across all test groups (Fig. 1I). These results indicate a critical role for NUCB2 in maintaining the integrity of intestinal glucose-sensing mechanisms, with its deficiency seemingly compromising this mechanism in rats.

To further investigate the influence of NUCB2/nesfatin-1 deficiency on intestinal lipid-sensing mechanisms, we conducted a lipid infusion experiment in the duodenum of NCD-fed WT and NUCB2-KO rats (Fig. 1J). Upon duodenal lipid infusion, we observed an increased GIR and decreased HGP in NCD-fed WT mice, suggesting inhibition of glucose production by intestinal lipid sensing under normal conditions (Fig. 1K–N). Conversely, lipid-infused NUCB2-KO rats presented a notable decrease in GIR (Fig. 1K,L) and a significant impairment in the suppression of HGP (Fig. 1M,N) compared to their WT counterparts. Glucose uptake remained unchanged across all groups (Fig. 1O). These results indicate that NUCB2 deficiency impairs intestinal nutrition-sensing mechanisms.

To investigate the effects of intestinal nesfatin-1 deficiency on intestinal nutrition-sensing mechanisms under IR conditions, we repeated gut glucose or lipid infusions in conjunction with the clamp technique in high-fat diet (HFD)-fed WT or NUCB2-KO rats (Fig. 2A,G). Compared with HFD-fed WT rats, HFD-fed NUCB2-KO rats presented reductions in GIR and HGP suppression, regardless of unaltered glucose uptake during both glucose and lipid infusion (Fig. 2B–F,H–L). Collectively, these results highlight the pivotal role of intestinal nesfatin-1 in regulating gut nutrient sensing and HGP under both standard and IR conditions.

## Effects of gut nesfatin-1 on hepatic glucose metabolism in NUCB2-KO rats

To investigate the effect of intestinal nesfatin-1 on glucose metabolism in vivo, a pilot study was conducted to determine the optimal concentration of nesfatin-1 for infusion (Fig. S2F). Our findings revealed a dose-dependent increase in the GIR when varying concentrations of nesfatin-1 protein were infused into the duodenum during PEC (Fig. S2G). Specifically, nesfatin-1 (100 μg/kg) infusion at a rate of 2 μg/kg/min resulted in an approximately twofold increase in GIR ($3.63 \pm 0.5$ vs. $2.19 \pm 0.5$ mg/kg/min, $p < 0.01$). To prevent an increase in circulating nesfatin-1 levels, a nesfatin-1 protein concentration of 100 μg/kg (2 μg/kg/min) was selected for all subsequent experiments. We further confirmed that the infusion rate did not affect the circulating levels of nesfatin-1, as determined by ELISA (Fig. S2H).

To address the role of intestinal nesfatin-1 in glucose homeostasis and insulin sensitivity in rats with different nutritional statuses, we performed a fasting-refeeding experiment during duodenal nesfatin-1 infusion in a body weight-matched cohort

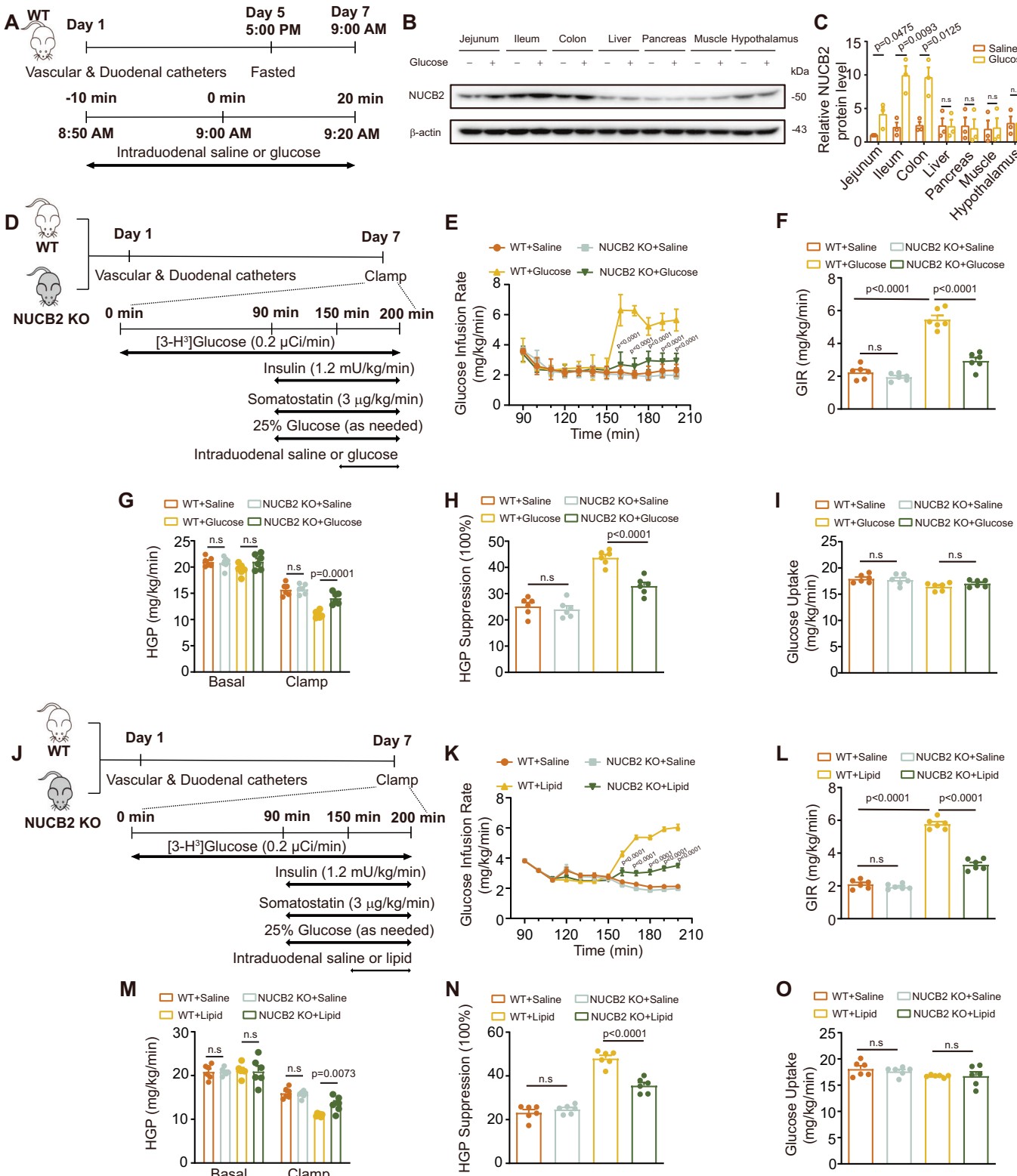

with NUCB2/nesfatin-1 deficiency (Fig. S3A). Upon fasting and subsequent refeeding, a rapid increase in blood glucose typically occurs, which is mitigated by reduced hepatic gluconeogenesis or increased glycogen synthesis in the liver (Duran-Sandoval et al, 2005). We hypothesized that duodenal nesfatin-1 may contribute to nutrient-sensing pathways triggered by refeeding, thereby further inhibiting HGP and reducing blood glucose. Indeed, we found that the blood glucose levels in NUCB2-KO rats with gut nesfatin-1 infusion were lower than those in animals with gut saline infusion 20 min after feeding (Fig. S3B). However, food intake remained

◄ **Figure 1. Effect of nesfatin-1 deficiency on intestinal nutrition-sensing mechanisms during duodenal glucose and lipid infusion in NCD-fed rats.**

(A) Experimental procedure for the effect of duodenal glucose infusion on nesfatin-1 expression in different tissues. (B, C) NUCB2/ nesfatin-1 protein expression (B) and quantitative analysis (C) in different tissues. (D) Experimental procedure and clamp protocol for duodenal glucose infusion in NCD-fed SD (WT) and NUCB KO rats. (E) Time course of GIR changes during the clamp. (F) Average GIR during the steady state of the clamp. (G) HGP. (H) Suppression of HGP. (I) Glucose uptake. (J) Experimental procedure and clamp protocol for duodenal lipid infusion in NCD-fed WT and NUCB2-KO rats. (K) Time course of GIR changes during the clamp. (L) Average GIR during the steady state of the clamp. (M) HGP. (N) Suppression of HGP. (O) Glucose uptake. NCD normal chow diet, PEC pancreatic-euglycemic clamp, GIR glucose infusion rates, HGP hepatic glucose production, n.s no significance. Values are shown as the mean ± SEM ($n = 3$–6 rats). Unpaired Student's *t*-test was used for (C), three-way ANOVA followed by Bonferroni's test was used for (E, K), and two-way ANOVA followed by Bonferroni's test was used for (F–I) and (L–O). Source data are available online for this figure.

comparable across both groups (Fig. S3C). These findings suggest that intestinal nesfatin-1 infusion improves the ability of animals to control postprandial blood glucose levels. We further explored this by activating duodenal nesfatin-1 signaling via duodenal nesfatin-1 infusion and subsequent glucose tolerance tests (GTTs) and insulin tolerance tests (ITTs) in NUCB2-KO rats fed an NCD. Compared with saline infusion, duodenal nesfatin-1 infusion reduced the area under the curve for both GTT ($AUC_{GTT}$) and ITT ($AUC_{ITT}$) (Fig. S3D,E), suggesting that duodenal nesfatin-1 infusion enhanced glucose tolerance and increased insulin response under conditions of systemic NUCB2/nesfatin-1 deficiency.

To delineate the effects of intestinal nesfatin-1 on hepatic glucose flux and whole-body insulin sensitivity in vivo, we performed a PEC experiment using NUCB2-KO rats that received either saline or nesfatin-1 infusions in the duodenum (Fig. 3A). Compared with saline infusion, duodenal administration of nesfatin-1 led to a marked elevation in the GIR (Fig. 3B,C), coupled with pronounced inhibition of HGP (Fig. 3D,E); however, glucose uptake was comparable in both groups (Fig. 3F). Collectively, these results revealed that duodenal nesfatin-1 enhances both glucose homeostasis and insulin sensitivity in vivo.

## GLP-1 signaling is needed for intestinal nesfatin-1-mediated regulation of HGP

Building on previous research indicating that nesfatin-1 stimulates the secretion of GLP-1 and GIP in intestinal secretory tumor (STC-1) cells (Ramesh et al, 2015), we proposed that intestinal nesfatin-1 can promote the secretion of GLP-1 by intestinal endocrine cells. Therefore, STC-1 cells and intestinal tissue fragments from NUCB2-KO rats were treated with nesfatin-1. As expected, nesfatin-1 protein treatment promoted GLP-1 secretion in small intestinal tissue slices (Fig. 3G) and STC-1 cells (Fig. 3H).

To further explore the potential role of GLP-1 signaling in the effect of intestinal nesfatin-1 on HGP in vivo, we co-infused exendin 9–39 (Ex-9), a GLP-1 receptor (GLP-1R) antagonist, and nesfatin-1 into the duodenum of NUCB2-KO rats (Fig. 3A). As expected, duodenal nesfatin-1 infusion promoted an increase in GIR and a decrease in HGP. Notably, the co-infusion of Ex-9 with nesfatin-1 largely abrogated these effects, indicating the critical involvement of GLP-1 signaling in mediating the effects of intestinal nesfatin-1 on GIR and HGP (Fig. 3B–E). No significant differences were observed in glucose uptake across all groups (Fig. 3F). Collectively, these in vivo and in vitro results underscore the possible interdependence between intestinal nesfatin-1 and GLP-1 signaling in the regulation of glucose metabolism. The regulatory influence of intestinal nesfatin-1 on HGP likely occur via

the enhancement of GLP-1 secretion or through direct interaction with GLP-1R.

## Associations between NUCB2/nesfatin-1 and MC4R-cAMP-GLP-1 signaling in vitro

Following our previous observation that Ex-9 significantly attenuates the effect of intestinal nesfatin-1 on HGP, we wondered whether nesfatin-1 can interact with GLP-1R. To address this question, we performed a co-immunoprecipitation (IP) experiment using mouse enteroendocrine STC-1 cells. Plasmids expressing NUCB2 (pc-NUCB2-Flag) and GLP-1R (pc-GLP-1R-HA) were successfully transfected into STC-1 cells, confirming the efficacy of the transfection process (Fig. S4A). However, the co-IP experiment did not reveal any interaction between nesfatin-1 and GLP-1R (Fig. S4B), indicating that nesfatin-1 does not directly bind to GLP-1R to exert its effects.

To uncover the molecular basis of how intestinal NUCB2/nesfatin-1 regulates HGP, we employed the Search Tool for the Retrieval of Interacting Genes/Proteins (STRING) database to establish a protein–protein interaction (PPI) network. This analysis identified 15 proteins within the immediate interaction network of NUCB2/nesfatin-1, notably including MC4R (Fig. 4A). Subsequent Kyoto Encyclopedia of Gene and Genomes (KEGG) pathway analysis revealed a strong correlation between NUCB2/nesfatin-1 PPIs and the cAMP signaling pathway (Fig. 4B). Ontology (GO) analysis molecular function (MF) analysis revealed the involvement of a G protein-coupled receptor (GPCR), MC4R, as a potential interacting protein of NUCB2/nesfatin-1 (Fig. S5A). In the biological process (BP) category, GO analysis revealed that NUCB2/nesfatin-1-related PPIs play a role in regulating intestinal transport, glucagon secretion, and feeding behavior, thus reinforcing their relevance to intestinal signaling in the regulation of glucose metabolism (Fig. S5B).

The role of nesfatin-1 has been shown to be related to MC4R (Rojczyk et al, 2015; Ying et al, 2015). In addition, a previous study reported that intestinal endocrine L cells express MC4R and regulate GLP-1 release in vivo (Panaro et al, 2014). In conjunction with our bioinformatic analysis, we hypothesized that intestinal nesfatin-1 promotes GLP-1 secretion to regulate HGP via an MC4R-dependent pathway. To address this hypothesis, STC-1 cells and intestinal tissue fragments from NUCB2-KO rats were treated with nesfatin-1 and/or the MC4R inhibitor SHU9119. Notably, SHU9119 treatment inhibited nesfatin-1-mediated GLP-1 secretion in intestinal tissue fragments from NUCB2-KO (Fig. 4C) and STC-1 cells (Fig. 4D). The ELISA results revealed a significant increase in GLP-1 levels in the culture medium of STC-1 cells and intestinal

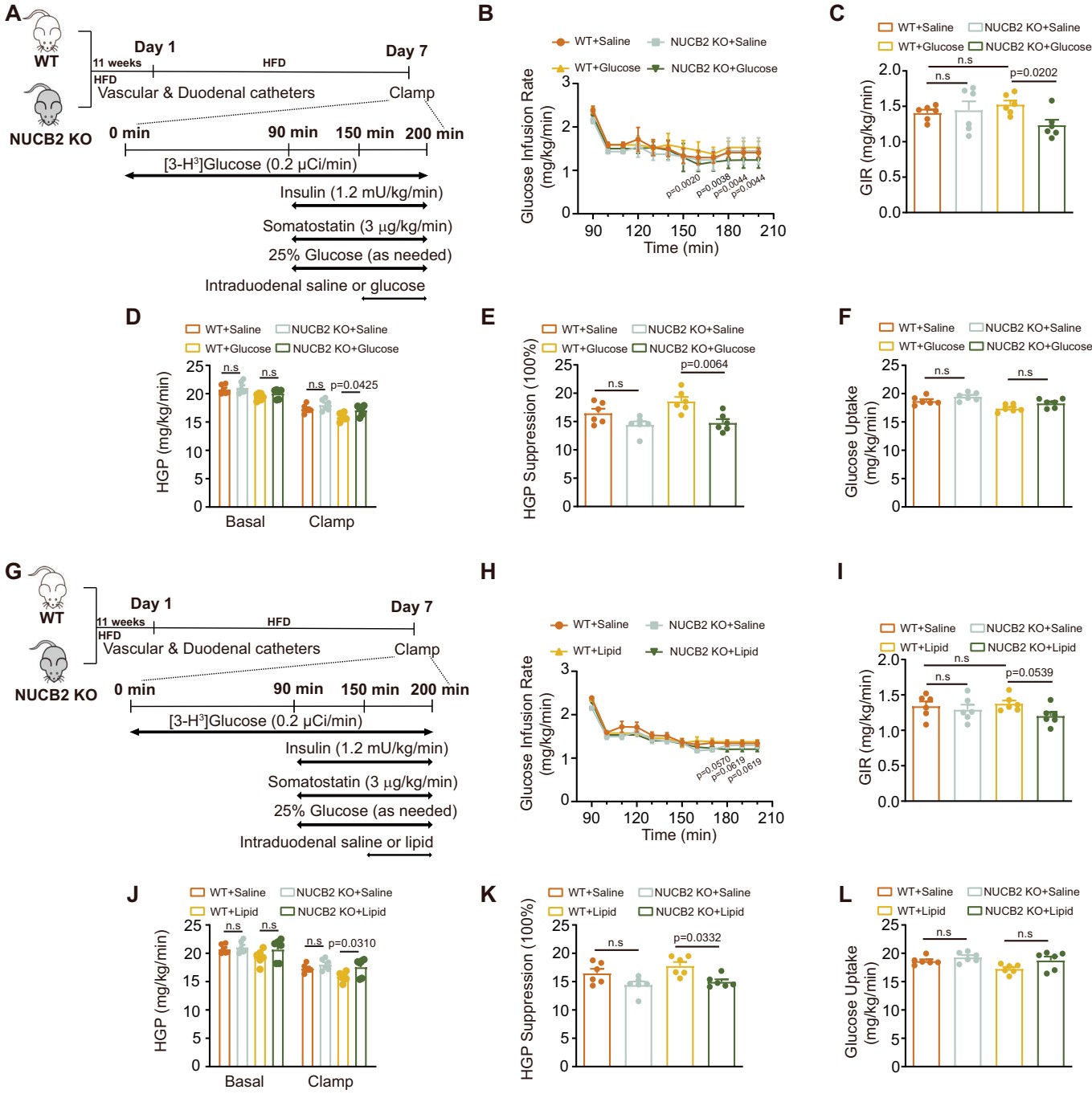

**Figure 2. Effect of nesfatin-1 deficiency on intestinal nutrition-sensing mechanisms during duodenal glucose and lipid infusion in HFD-fed rats.**

(**A**) Experimental procedure and clamp protocol for duodenal glucose infusion in HFD-fed WT and NUCB2-KO rats. (**B**) Time course of GIR changes during the clamp. (**C**) Average GIR during the steady state of the clamp. (**D**) HGP. (**E**) Suppression of HGP. (**F**) Glucose uptake. (**G**) Experimental procedure and clamp protocol for duodenal lipid infusion in HFD-fed WT and NUCB2-KO rats. (**H**) Time course of GIR changes during the clamp. (**I**) Average GIR during the steady state of the clamp. (**J**) HGP. (**K**) Suppression of HGP. (**L**) Glucose uptake. HFD high-fat diet, PEC pancreatic-euglycemic clamp, GIR glucose infusion rates, HGP hepatic glucose production, n.s no significance. Values are shown as the mean ± SEM ($n = 6$ rats). Three-way ANOVA followed by Uncorrected Fisher's LSD test was used for (**B, H**), two-way ANOVA followed by Uncorrected Fisher's LSD test was used for (**C, I**), and two-way ANOVA followed by Bonferroni's test was used for (**D–F**) and (**J–L**). Source data are available online for this figure.

tissue fragments from NUCB2-KO rats following nesfatin-1 treatment. This effect was abolished by SHU9119 (Fig. 4E,F), while SHU9119 alone did not affect the GLP-1 concentration. These results indicate that nesfatin-1 potentially facilitates GLP-1 secretion via an MC4R-mediated pathway, although the exact

interaction between nesfatin-1 and MC4R requires further investigation.

Given that KEGG analysis showed that the interacting protein of NUCB2/nesfatin-1 was related to cAMP signaling and that cAMP is downstream of MC4R(Anton et al, 2022; Molden et al, 2015), we

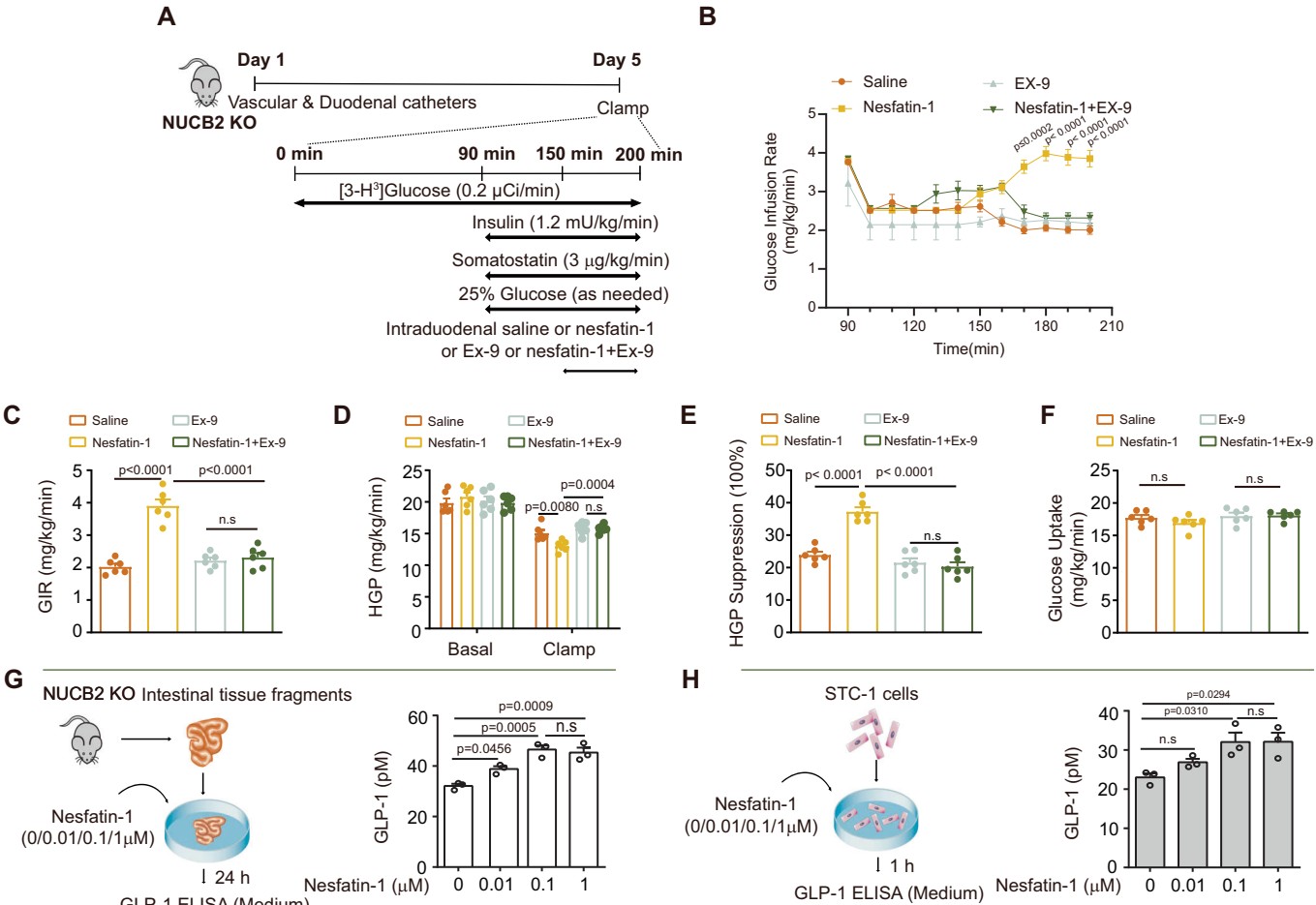

**Figure 3. GLP-1 signaling is needed for intestinal nesfatin-1 to regulate HGP.**

(A) Experimental procedure and clamp protocol. Eight-week-old male NUCB2-KO rats underwent duodenal, internal jugular vein and carotid artery catheterization on day 1. PECs were performed on day 5. Saline, nesfatin-1, Ex-9, or nesfatin-1 and Ex-9 were infused through a duodenal catheter. (B) Time course of the GIR changes during the clamp. (C) Average GIR during the steady state of the clamp. (D) HGP. (E) Suppression of HGP. (F) Glucose uptake. (G, H) Intestinal tissue fragments from NUCB2-KO mice or STC-1 cells were cultured and treated with different concentrations of nesfatin-1. The GLP-1 concentration in the culture medium was determined by ELISA in intestinal tissue fragments (G) and STC-1 cells (H). Ex-9, a GLP-1 receptor inhibitor; n.s no significance. Values are shown as the mean ± SEM ($n = 6$ rats or 3 independent experiments); Two-way ANOVA followed by Bonferroni's test was used for (B), one-way ANOVA followed by Bonferroni's test was used for (C–F), and one-way ANOVA followed by Tukey's test was used for (G, H). Source data are available online for this figure.

investigated the effect of nesfatin-1 on cAMP release in vitro. The ELISA results showed that the cAMP concentration in the culture medium of STC-1 cells and intestinal tissue fragments from NUCB2-KO rats was significantly increased after treatment with nesfatin-1 (Fig. 4E,F). However, this effect was eliminated by SHU9119 (Fig. 4E,F), suggesting a potential role of intestinal nesfatin-1 in modulating MC4R activity and the consequent increase in intracellular cAMP levels.

To further validate the aforementioned results, we used self-complementary adeno-associated virus (scAAV) to suppress the expression of MC4R in the rat gut and observed changes in cAMP and GLP-1 levels under nesfatin-1 stimulation (Fig. S6A). As expected, both MC4R mRNA and protein expression were significantly reduced in the intestinal tissue (Fig. S6B,C). Consistent with the aforementioned findings, cAMP and GLP-1 levels increased significantly under nesfatin-1 treatment, while MC4R inhibition abolished this effect of nesfatin-1 (Fig. S6D,E). In

addition, to confirm whether MC4R mediates the secretion of cAMP and GLP-1 stimulated by nesfatin-1 in vitro, STC-1 cells were transfected with adenovirus overexpressing or knockdown MC4R (Ad-*MC4R/shMC4R*), or with a control (Ad-*mCherry*). As expected, MC4R expression significantly increased or decreased in cells transfected with Ad-*MC4R/shMC4R* (Fig. S6F–H). After treatment with nesfatin-1, the changes in cAMP and GLP-1 levels in the medium were similar to those observed in the intestinal tissue fragments (Fig. S6I,J). Taken together, these results emphasize the important role of MC4R in the secretion of cAMP and GLP-1 stimulated by nesfatin-1.

## Intestinal nesfatin-1 regulates HGP through the MC4R-cAMP-GLP-1 signaling in vivo

To further investigate whether MC4R is crucial for the effect of intestinal nesfatin-1 on HGP, we conducted a series of experiments

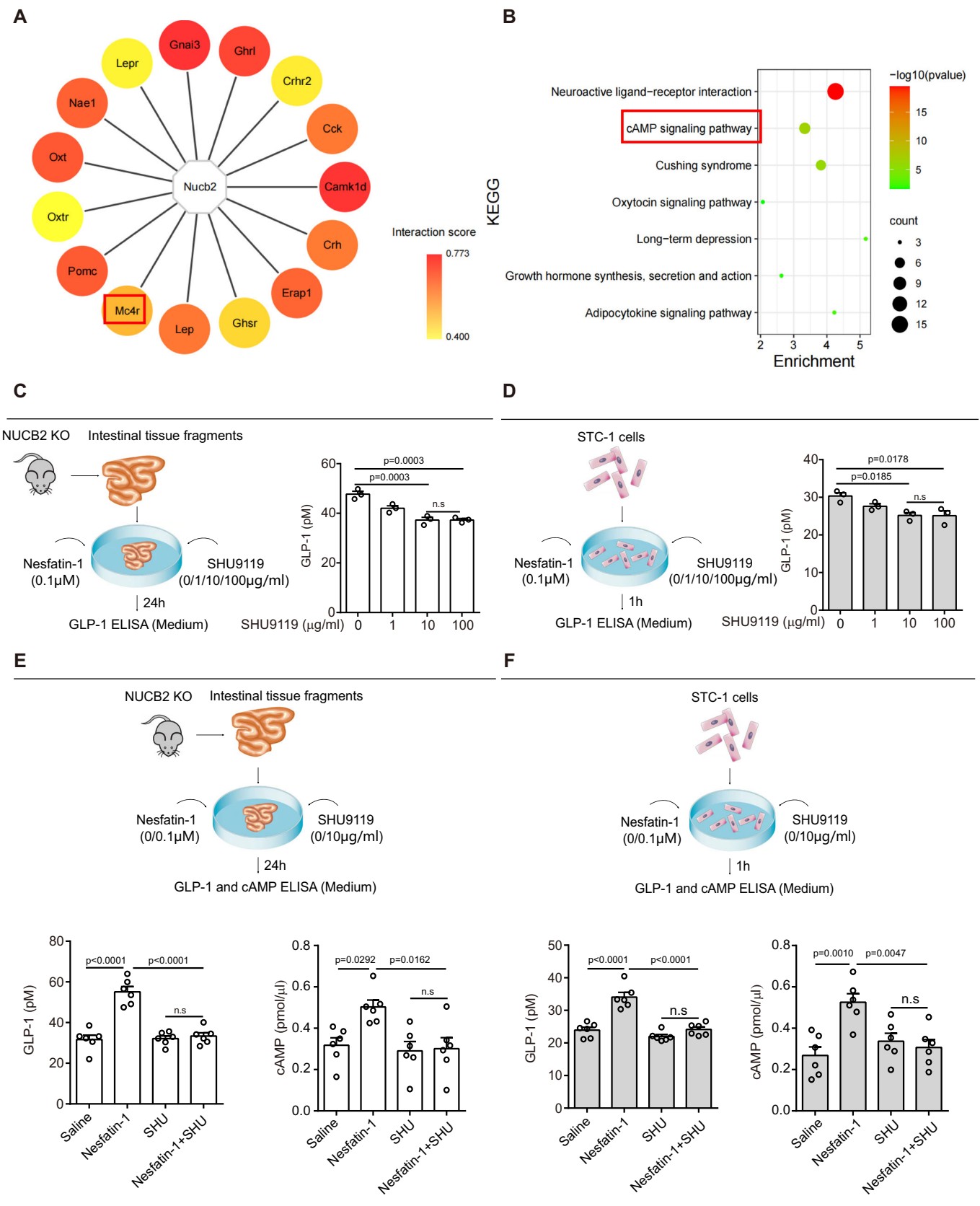

**Figure 4.    Association of nesfatin-1 with the MC4R-cAMP signaling pathway in vitro.**

(A) Protein–protein interaction (PPI) network of NUCB2. (B) Bubble plot of enriched KEGG pathway statistics. The color and size of the dots represent the range of −log10 and the number of genes mapped to the indicated pathways, respectively. Seven enrichment pathways are shown. (C, D) Small intestinal fragments of NUCB2-KO mice and STC-1 cells were cultured and treated with nesfatin-1 and different concentrations of SHU9119. GLP-1 levels in the culture medium of small intestinal fragments (C) and STC-1 cells (D) were determined by ELISA. (E, F) Small intestine fragments of NUCB2-KO mice and STC-1 cells were cultured and treated with and without nesfatin-1 or/ and SHU9119 (0 and 10 μg/ml). The cAMP (right) and GLP-1 (left) concentration in the culture medium of small intestinal fragments (E) and STC-1 cells (F) was determined by ELISA. n.s no significance. Values are shown as the mean ± SEM (*n* = 3 rats or 3 independent experiments). One-way ANOVA followed by Tukey's test was used for statistical analysis. Source data are available online for this figure.

in 8-week-old male NUCB2-KO rats. PECs were performed with duodenal nesfatin-1 and/or SHU9119 infusions through a catheter (Fig. 5A). Immunohistochemical (IHC) staining of the intestinal mucosa revealed a notable increase in the expression levels of both nesfatin-1 and MC4R in the duodenal epithelium following duodenal nesfatin-1 infusion compared with the saline control (Fig. 5B–D). Similarly, immunofluorescence (IF) images obtained by confocal microscopy revealed increases in both MC4R (red) and GLP-1 (green) expression in the intestinal epithelium after duodenal nesfatin-1 infusion, with nuclei stained with DAPI (blue) (Fig. 5E–G). However, the upregulated expression of MC4R and GLP-1 induced by intestinal nesfatin-1 was substantially reduced when co-infused with SHU9119 (Fig. 5E–G). Moreover, co-infusion of nesfatin-1 and SHU9119 largely abrogated the effects of intestinal nesfatin-1 on GIR (Fig. 5H,I) and HGP (Fig. 5J,K) in NUCB2-KO rats, whereas glucose uptake remained unchanged (Fig. 5L). Collectively, these findings provide compelling evidence that MC4R activation within the intestinal epithelium underpins the regulatory role of intestinal nesfatin-1 in glucose metabolism.

Building on the effects of nesfatin-1 and SHU9119 infusion, we sought to extend our understanding of the impact of gut nesfatin-1 on hepatic gluconeogenesis and insulin signaling at the molecular level. Through western blot analysis, we quantified the expression of pivotal proteins associated with gluconeogenesis and insulin signaling in hepatic tissue. In alignment with the outcomes of the PEC study, we observed significant decreases in the mRNA and protein expression of hepatic phosphoenolpyruvate carboxykinase (PEPCK) and glucose-6-phosphatase (G6Pase), two key enzymes involved in gluconeogenesis, in NUCB2-KO rats following duodenal nesfatin-1 infusion (Fig. 5M,N). Concurrently, there were increases in the phosphorylation of insulin receptor substrate-1 (IRS-1; Ser 302), insulin receptor (InsR; Tyr 1150/1151), and protein kinase B (Akt; Ser 473) in the liver (Fig. 5O). These results suggest inhibition of gluconeogenesis and enhancement of insulin sensitivity in the livers of NUCB2-KO rats upon duodenal nesfatin-1 infusion. Conversely, co-infusion of nesfatin-1 and SHU9119 in the intestine reversed these effects (Fig. 5H–O). Furthermore, in agreement with previous IHC findings, western blotting revealed that intestinal nesfatin-1 infusion increased the expression levels of nesfatin-1 and MC4R in the intestinal tissues of NUCB2-KO rats. However, the effect of duodenal nesfatin-1 on MC4R expression was abolished when nesfatin-1 and SHU9119 were simultaneously infused (Fig. 5P). These in vivo results collectively suggest that gut-derived nesfatin-1 may modulate HGP by interacting with MC4R, thereby facilitating GLP-1 secretion. The inhibitory effect of the MC4R antagonist SHU9119 suggested its role in obstructing the interaction between nesfatin-1 and MC4R.

## Interaction between nesfatin-1 and MC4R in vitro

To identify the proteins that interact with nesfatin-1, IP and MS analyses were performed using pc-*NUCB2*-Flag-transfected STC-1 cells (Fig. 6A). MS analysis confirmed the presence of MC4R in NUCB2-Flag IP (Fig. 6B), suggesting a possible interaction between NUCB2/nesfatin-1 and MC4R. This hypothesis was further supported by confocal microscopy, which revealed the co-localization of NUCB2/ nesfatin-1 and MC4R within small intestinal mucosal cells (Fig. 6C). To validate this PPI, we conducted co-IP experiments to verify this interplay at the protein level by transfecting STC-1 cells with plasmids expressing NUCB2 (pc*NUCB2*-Flag) and/or MC4R (pc*MC4R*-HA) (Fig. S4A). The subsequent co-IP results revealed that NUCB2/ nesfatin-1 and MC4R formed a complex in STC-1 cells (Fig. 6D,E). Furthermore, we found that endogenous NUCB2/nesfatin-1 interacted with MC4R and vice versa (Fig. 6F). To further investigate the nature of this interaction, we employed a fluorescence resonance energy transfer (FRET) assay. The FRET assay further substantiated the molecular-level interaction between NUCB2/nesfatin-1 and MC4R (Fig. 6G–J), providing compelling evidence for the direct interplay between these two proteins.

Building on this verified interaction, we sought to identify the specific domain within NUCB2 that engages in MC4R. NUCB2 has a pointed nesfatin-1 domain (25–106 amino acids [aa]), a DNA-binding domain (171–223 aa), and an EF-hand domain (247-322 aa). To discern the region involved in MC4R binding, we constructed Flag-tagged WT NUCB2 and several deletion mutant plasmids, which contained three deletion mutations (ΔN25–50 aa, ΔN51–75 aa, and ΔN76–106 aa) in NUCB2/nesfatin-1 (Fig. 6K). Upon assessment, we found that the absence of the 51–75 aa domain resulted in a loss of MC4R binding, whereas deletion of the other regions had no discernible impact on the NUCB2/nesfatin-1 to MC4R interaction (Fig. 6L). Therefore, we believe that the 51–75 aa region, which is a segment of the NUCB2/nesfatin-1 sequence, is crucial for the binding interaction between NUCB2 and MC4R.

To further explore the core site of the interaction between NUCB2 and MC4R, we compared the amino acid sequence of NUCB2/nesfatin-1 (51–75 aa) with those of α-melanocyte-stimulating hormone (α-MSH) and γ-melanocyte-stimulating hormone (γ-MSH), both of which are recognized as endogenous agonists of MC4R. The results revealed the presence of a shared amino acid sequence, $Phe^7$-$Arg^8$-$Trp^9$ ($_7$H-F-$R_9$) (Fig. 6M). This is of interest, as it has been established that the smallest domain of MC4R that binds to ligands is Phe-Arg-Trp (Yang and Harmon, 2017). Therefore, we hypothesized that the $_{70}$H-F-$R_{72}$ sequence within the 51–75 aa span of NUCB2/nesfatin-1 potentially constitutes the MC4R binding site.

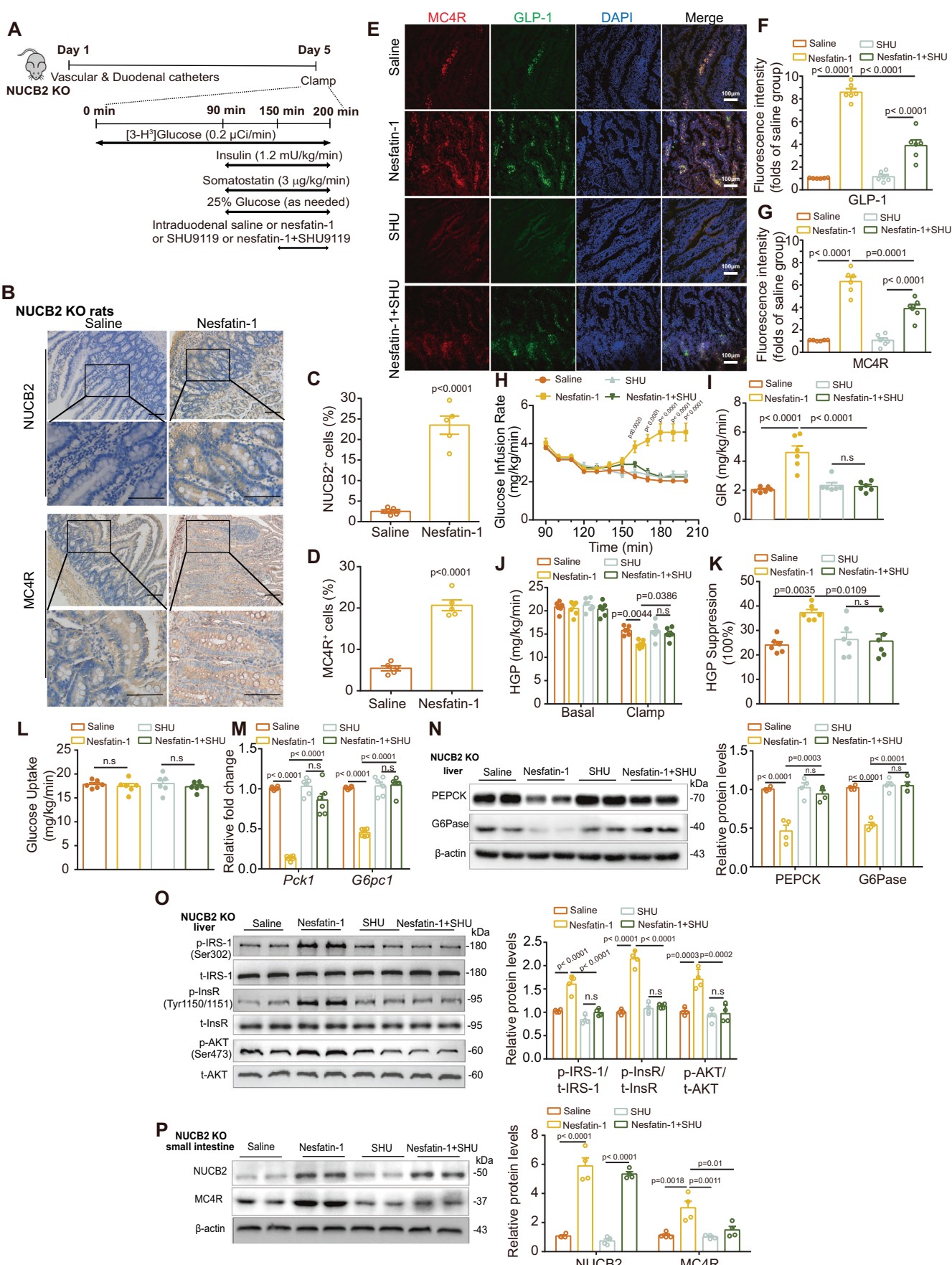

**Figure 5.  MC4R-cAMP-GLP-1 signaling mediates the effect of intestinal nesfatin-1 on HGP in vivo.**

(A) Experimental procedure and clamp protocol. Eight-week-old male NUCB2-KO rats underwent duodenal, internal jugular vein and carotid artery catheterization on day 1. PECs were performed on day 5. Nesfatin-1 and/or SHU9119 were infused through a duodenal catheter. (B) IHC images for NUCB2 and MC4R expression in the intestinal mucosa (Scale bars, 100 μm). (C) Quantitative analysis of NUCB2$^+$ cells. (D) Quantitative analysis of MC4R$^+$ cells. (E) IF images for GLP-1 and MC4R expression in the intestinal mucosa (Scale bars, 100 μm). (F, G) Integrated density quantification of GLP-1 (F) and MC4R (G). (H) Time course of GIR during the clamp. (I) Average GIR during the steady state of the clamp. (J) HGP. (K) Suppression of HGP. (L) Glucose uptake. (M) *Pck1* and *G6pc1* mRNA expression in the liver. (N) G6Pase and PEPCK protein expression in the liver (left) and density quantification (right). (O) t-IRS-1/p-IRS-1 (Ser 302), t-InsR/p-InsR (Tyr 1150/1151) and t-Akt/p-Akt (Ser 473) (left), and density quantification (right). (P) NUCB2 and MC4R protein expression in the intestinal mucosa (left) and density quantification (right). n.s no significance. Values are shown as the mean ± SEM (*n* = 4–6 rats). Unpaired Student's *t*-test was used for (C, D), Two-way ANOVA followed by Bonferroni's test was used for (H), and one-way ANOVA followed by Bonferroni's test was used for (F, G) and (I–P). Source data are available online for this figure.

To achieve a stable MC4R/NUCB2 complex, we used the root mean square deviation (RMSD) to monitor the molecular dynamics (MD) simulation for 100 ns. The RMSD plots showed that MC4R stabilized at ~3.0 Å after 12 ns (Fig. 6N,O). NUCB2_HFR had a larger shift in the period of the MD simulation, and the fluctuation stabilized at ~5.0 Å from 40 ns (Fig. 6N). However, the RMSD for MC4R and NUCB2_AAA (wherein the NUCB2 $_{70}$H-F-R$_{72}$ segment was mutated to $_{70}$A-A-A$_{72}$) stabilized at 3.0 and 2.5 Å, respectively, for the MC4R/NUCB2_AAA mutant complex, indicating the stability of both complexes (Fig. 6O). Co-IP results showed that when $_{70}$H-F-R$_{72}$ of NUCB2/nesfatin-1 was mutated to $_{70}$A-A-A$_{72}$ (NUCB2$^{\Delta HFR}$-Flag), the interaction between NUCB2/nesfatin-1 and MC4R was eliminated, in contrast to the unmutated NUCB2-Flag (Fig. 6P), indicating that the $_{70}$H-F-R$_{72}$ segment in NUCB2/nesfatin-1 is indeed the MC4R interaction site.

Finally, we analyzed the interaction energy between MC4R and NUCB2. The alignment showed that NUCB2_HFR (blue) could be inserted into the binding groove more profoundly than NUCB2_AAA (orange) (Fig. 6Q,R). Specifically, residue Arg72 of NUCB2$^{WT}$ demonstrated the potential to form two salt bridges with Glu100 and Asp126 of MC4R; the distances were 2.9–3.5 Å and 2.8–3.0 Å, respectively, and the interaction energies were −17.0199 and −18.6399 kcal/mol, respectively. In addition, Arg72 could bind to Val119 of MC4R through hydrogen bonding (~3.7 Å) and contribute −2.1780 kcal/mol of interaction energy to the MC4R/NUCB2_HFR complex. We found that His70 of NUCB2_HFR formed a weak hydrogen bond (~3.5 Å, −3.0090 kcal/mol) with His264 of MC4R (Fig. 6Q). However, NUCB2_AAA and MC4R did not exhibit such interactions (Fig. 6R). Taken together, these data reaffirmed our hypothesis that NUCB2/nesfatin-1 interacts with MC4R via its $_{70}$H-F-R$_{72}$ sequence.

## The $_{70}$H-F-R$_{72}$ domain of NUCB2/nesfatin-1 is key for the regulation of HGP by intestinal nesfatin-1

To verify the effect of NUCB2$^{\Delta HFR}$ mutation in vivo, we infused mutant nesfatin-1 protein (nesfatin-1$^{\Delta HFR}$) or WT nesfatin-1 protein (nesfatin-1$^{WT}$) into the duodenum and performed PEC in NUCB2- KO rats (Fig. 7A). Compared with saline infusion, nesfatin-1$^{WT}$ infusion significantly increased GIR (Fig. 7B, C) and inhibited HGP (Fig. 7D,E) in NUCB2-KO rats, whereas glucose uptake remained unchanged (Fig. 7F). In contrast, duodenal nesfatin-1$^{\Delta HFR}$ infusion did not produce discernible changes in the GIR or HGP in NUCB2-KO rats (Fig. 7B–E). Consistent with the results of the PEC study, duodenal nesfatin-1$^{\Delta HFR}$ infusion did not result in a considerable decrease in hepatic PEPCK or G6Pase protein expression in NUCB2-KO rats (Fig. 7G). In addition, duodenal nesfatin-1$^{\Delta HFR}$ infusion failed to increase the

phosphorylation of hepatic IRS-1 (Ser 302), InsR (Tyr 1150/1151), or Akt (Ser 473) in these rats (Fig. 7H). Notably, Western blotting revealed that intestinal infusion of nesfatin-1$^{\Delta HFR}$ increased NUCB2 expression but had no effect on MC4R expression in the intestinal tissues of NUCB2-KO rats (Fig. 7I). These findings collectively underscore the importance of the $_{70}$H-F-R$_{72}$ site of NUCB2/nesfatin-1 in the regulation of HGP by intestinal nesfatin-1. Next, we examined the effects of nesfatin-1$^{\Delta HFR}$ on MC4R-cAMP signaling and GLP-1 secretion in vitro. Consistent with our in vivo findings, western blotting showed that nesfatin-1$^{wt}$ increased MC4R expression in both intestinal fragments of NUCB2-KO rats and STC-1 cells, whereas nesfatin-1$^{\Delta HFR}$ had no significant effect (Fig. 7J,K). Furthermore, nesfatin-1$^{wt}$ treatment increased both cAMP and GLP-1 levels in the culture medium of intestinal tissue fragments and STC-1 cells, whereas nesfatin-1 $^{\Delta HFR}$ treatment failed to induce any notable changes in these parameters (Fig. 7J,K). Collectively, our data confirmed that the $_{70}$H-F-R$_{72}$ domain of NUCB2/nesfatin-1 is the pivotal site through which nesfatin-1 promotes GLP-1 secretion and orchestrates regulatory control over HGP in vivo.

## Duodenal nesfatin-1 reduces HGP via the gut–brain–liver neural circuit

Building on our previous observations, we next sought to determine whether the inhibitory influence of intestinal nesfatin-1 on HGP involves the activation of gut-brain-liver neurocircuitry, similar to the mechanism by which duodenal GLP-1 regulates HGP (Yang et al, 2017) (Fig. S7A,B). We infused nesfatin-1 with or without tetracaine, a neurotransmitter inhibitor (Wang et al, 2008), into the duodenum of NUCB2-KO rats. We found that intestinal infusion of tetracaine alone had no discernible effects on GIR or HGP, but the co-infusion of tetracaine and nesfatin-1 into the duodenum completely abrogated the nesfatin-1-induced enhancement of GIR and suppression of HGP (Fig. S7C–E). Glucose uptake was not affected in any group (Fig. S7F). These data underscore the indispensability of the intestinal vagal nerve for HGP regulation by nesfatin-1.

The afferent vagus nerve terminates in the nucleus of the solitary tract (NTS) within the dorsal vagal complex (DVC). The N-methyl-D-aspartate (NMDA) receptor is located in neurons of the NTS in the posterior brain and mediates peripheral signals induced by hormones to regulate food intake and energy metabolism (Wright et al, 2011). Therefore, to assess whether NMDA receptors are essential for intestinal nesfatin-1-mediated neuronal transmission to reduce HGP, we infused the NMDA receptor antagonist, MK-801, into the NTS of NUCB2-KO rats during PEC (Wang et al,

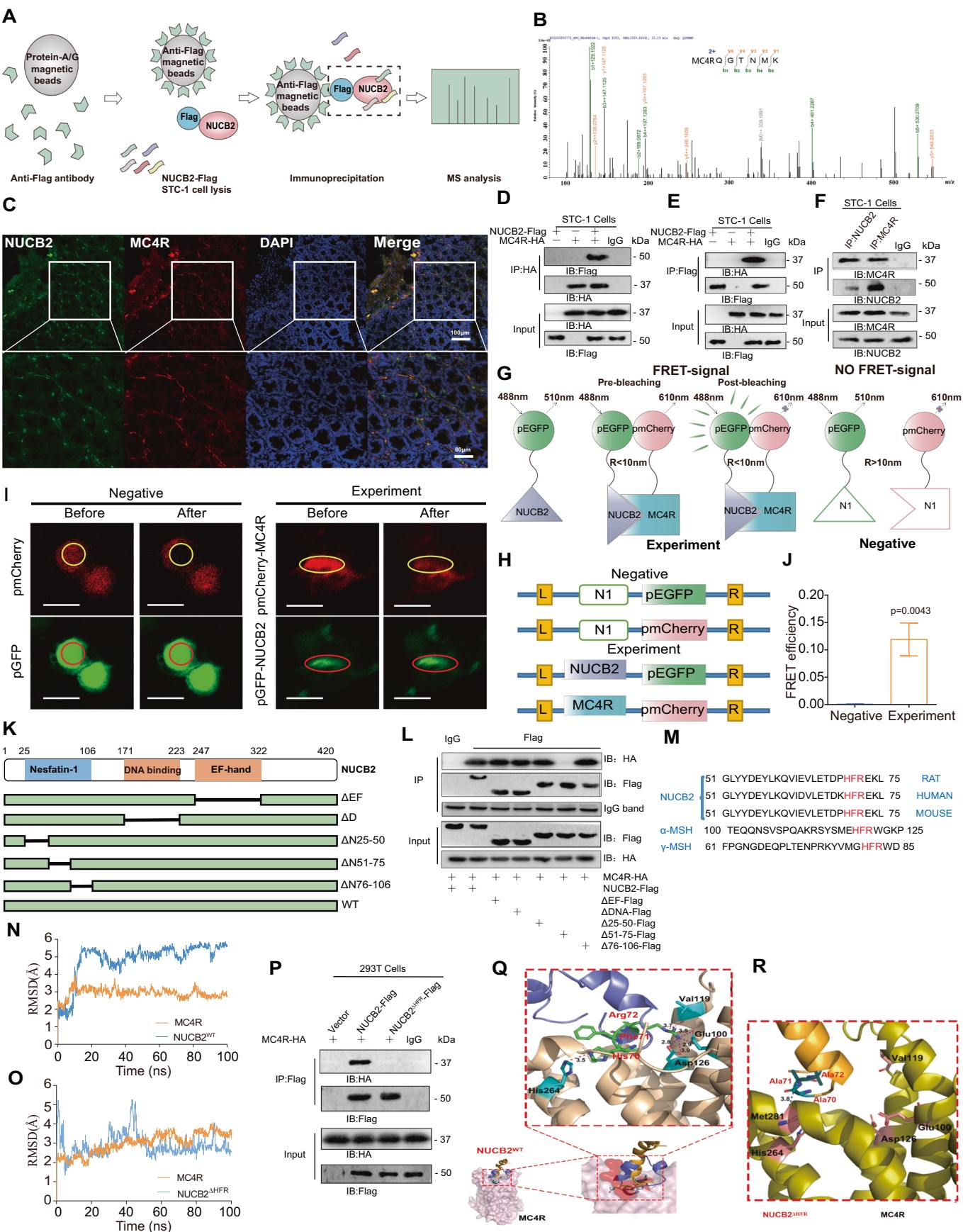

◀

**Figure 6.  Interaction between nesfatin-1 and MC4R is needed for the regulation of HGP by gut nesfatin-1.**

(A) The experimental design diagram for the IP–LC–MS analysis. (B) LC–MS analysis of NUCB2-Flag identified MC4R. (C) NUCB2 and MC4R were co-localized in the small intestinal tissue. Images were taken with a confocal microscope (scale bars, 60 and 100 μm). (D, E) Co-IP experiments of MC4R-HA and NUCB2-Flag were performed using an anti-HA antibody (D) or an anti-Flag antibody (E) in STC-1 cells. (F) Co-IP of endogenous MC4R and NUCB2 in STC-1 cells. (G) Schematic representation of the FRET assay in vitro, a system for two-component protein interaction. GFP-NUCB2 was excited by fluorescence light at a wavelength of 488 nm with a wavelength bandwidth of 10 nm (488–10 nm). When excited, it emits longer excitation wavelengths of 510–10 nm. When NUCB2 interacts with MC4R, the receptor (mCherry) approaches GFP and generates a 610–10 nm FRET signal between the two fluorescent proteins. GFP fluorescence is enhanced after mCherry fluorescence bleaching. When the two fluorescent proteins are far apart, the above phenomenon disappears. R is the distance between two fluorescent proteins. (H) Schematic diagram of plasmid construction for the FRET assay. Two fluorescent proteins, GFP and mCherry, are attached to the C-terminus of the NUCB2 and MCR4 proteins, respectively. (I, J) Fluorescence imaging (I) and efficiency (J) of the FRET assay for the negative or experimental plasmids in 293T cells. The circle represents the fluorescent bleaching area (scale bar, 20 μm). (K) Schematic diagram of the NUCB2 mutant. (L) A series of Flag-tagged NUCB2 deletion mutants were transfected into STC-1 cells for 48 h, followed by co-IP assays to test the interactions between different domains of NUCB2 and MC4R. (M) Sequence alignment of NUCB2, α-MSH, and γ-MSH. (N) RMSD plot for MC4R and NUCB2_HFR. (O) RMSD plots for MC4R and NUCB2_AAA. (P) 293T cells were transfected with vectors encoding Flag-tagged versions of NUCB2 or NUCB2$^{\Delta HFR}$. IP with anti-flag magnetic beads was performed in cell lysates. (Q, R) Binding mode between MC4R and NUCB2. (Q) The interaction between MC4R and NUCB2_HFR. (R) The interaction between MC4R and NUCB2_AAA. FRET fluorescence resonance energy transfer, n.s no significance. Values are shown as the mean ± SEM (n = 3 independent experiments). Unpaired Student's t-test was used for statistical analysis. Source data are available online for this figure.

2008) (Fig. S7A,B). MK-801 effectively eliminated nesfatin-1-triggered increases in GIR and the suppression of HGP (Fig. S7C–E), with no observable effects on glucose uptake (Fig. S7F), indicating the integral role of NTS NMDA receptors in the modulation of HGP by intestinal nesfatin-1.

Finally, to explore the pathway by which duodenal nesfatin-1 regulates HGP, we analyzed the effect of hepatic branch vagotomy (HBV) on intestinal nesfatin-1-mediated inhibition of HGP (Fig. S7A,B). HBV completely eliminated the enhancement of GIR and the reduction in HGP stimulated by duodenal nesfatin-1 (Fig. S7C–E) without affecting glucose uptake (Fig. S7F). This finding implicates the hepatic branch of the efferent vagal nerve as the means through which the NTS transmits gut nesfatin-1 signaling to the liver to modulate HGP. Taken together, our findings provide compelling evidence that duodenal nesfatin-1 orchestrates HGP through the gut-brain-liver neural circuit.

# Discussion

NUCB2/nesfatin-1 has garnered considerable attention for its role in regulating energy homeostasis and glucose and lipid metabolism in the central nervous system (CNS) and peripheral tissues (Wu et al, 2014; Yang et al, 2012). However, despite its predominant expression in the GI tract, the precise function of NUCB2/nesfatin-1 remains unknown. In this study, we found that NUCB2/nesfatin-1 plays a crucial role in maintaining the integrity of the intestinal nutrient-sensing mechanism in rats. Importantly, we demonstrated that a duodenal nesfatin-1 infusion in NUCB2-KO rats led to significant improvements in glucose homeostasis, including the inhibition of HGP and increased insulin sensitivity. These effects are mediated by the activation of the MC4R-cAMP-GLP-1 signaling pathway, highlighting the importance of this molecular pathway in the regulation of glucose metabolism. Furthermore, we demonstrated the involvement of the gut-brain-liver neural circuitry in mediating the regulatory signals of nesfatin-1. These findings highlight the important role of gut nesfatin-1 in the regulation of hepatic glucose metabolism and insulin response, suggesting that nesfatin-1 may be a promising novel therapeutic intervention for metabolic disorders.

Herein, we first investigated the response of NUCB2/nesfatin-1 to changes in intestinal nutritional status through intestinal glucose

infusion in WT rats, and found that NUCB2 expression was significantly increased in the intestine, indicating increased synthesis and secretion of nesfatin-1. Similarly, previous studies have reported that changes in gut nutrient concentrations lead to alterations in the secretion of gut hormones such as GLP-1 and glucose-dependent insulinotropic polypeptide (GIP) (Bauer et al, 2018; Lu et al, 2007; Wu et al, 2015). Furthermore, enteroendocrine STC-1 cells increased the protein expression of metabolism-related genes after short-term glucose stimulation. Taken together, these reports indicate that the response of the gut to changes in nutritional signals is rapid and sensitive. However, the mechanism of this reaction remains unknown. We speculate that changes in nutritional signals may trigger intestinal nutrient-sensing mechanisms, leading to alterations in the expression and secretion of metabolic factors (Breen et al, 2013).

Although previous studies have explored the effects of intestinal hormones on nutrient sensing and identified several hormones that regulate HGP (Martin et al, 2019), one major challenge in studying intestinal hormones is the exclusion of the influence of endogenous hormones. This limitation hinders accurate assessment of the true effects of intestinal hormones on hepatic glucose metabolism. To address these issues, we constructed NUCB2-KO rats and employed a PEC to suppress endogenous insulin via the infusion of somatostatin, thereby maintaining blood glucose and insulin levels close to baseline in each group of rats. On this basis, we used GIR to determine the differences in insulin sensitivity among different groups of rats. In HFD-fed rats, due to the infusion of somatostatin inhibiting their hyperinsulinemia, insulin levels during the clamp steady-state were low and close to the baseline levels of control rats. In addition, both somatostatin-induced inhibition of glucagon and exogenous glucose infusion have inhibitory effects on HGP (Abraham et al, 2018; Cherrington, 1999; Kowalchuk et al, 2017). Therefore, under conditions of NUCB2/nesfatin-1 deficiency and PEC, we studied the role of duodenal nesfatin-1 infusion on HGP and elucidated the underlying mechanism for the first time. Our results revealed impaired nutrient-sensing mechanisms within the gut-brain-liver neural network in NUCB2-KO rats, leading to the absence of HGP inhibition upon duodenal glucose or lipid infusion, which is typically observed under normal conditions. Furthermore, our findings indicated that even in the absence of endogenous NUCB2/nesfatin-1, the accumulation of nesfatin-1 in the intestine is

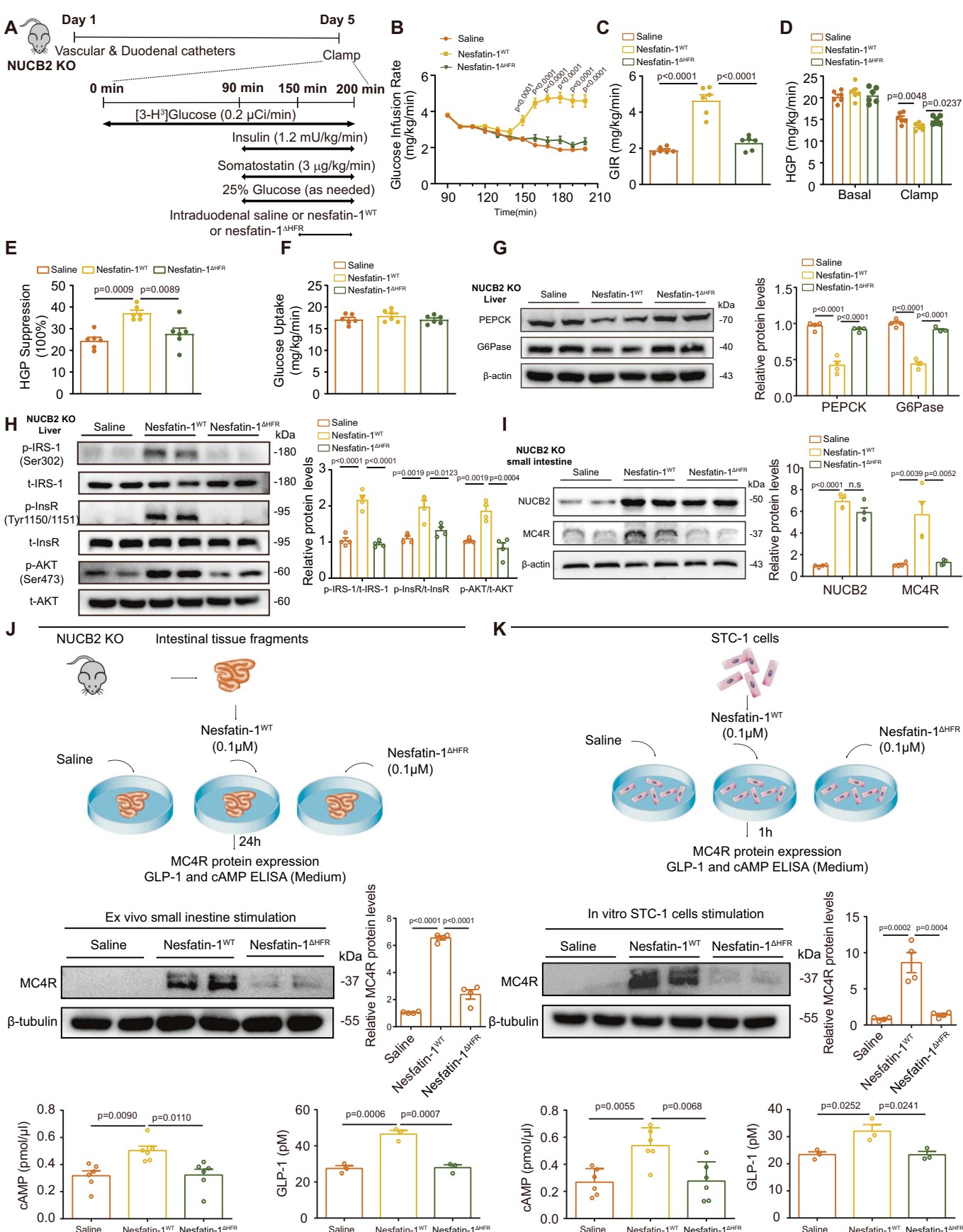

**Figure 7.  The HFR domain of NUCB2 is pivotal for the regulation of HGP by intestinal nesfatin-1.**

(A) Experimental procedure and clamp protocol. Eight-week-old male NUCB2-KO rats underwent duodenal, internal jugular vein, and carotid artery catheterization on day 1. PECs were performed on day 5. Saline, Nesfatin-1$^{WT}$, or Nesfatin-1$^{\Delta HFR}$ protein was infused through a duodenal catheter. (B) Time course of the GIR changes during the clamp. (C) Average GIR during the steady state of the clamp. (D) HGP. (E) Suppression of HGP. (F) Glucose uptake. (G) G6Pase and PEPCK protein expression (left) and density quantification (right) in the liver. (H) Hepatic t-IRS-1/p-IRS-1(Ser 302), t-InsR/p-InsR (Tyr 1150/1151), and t-Akt/p-Akt (Ser 473) levels (left) and density quantification (right). (I) Protein expression of NUCB2 and MC4R in intestinal tissue fragments (left) and density quantification (right). (J) Small intestine fragments of NUCB2-KO rats were cultured and treated with saline, nesfatin-1$^{WT}$, or nesfatin-1$^{\Delta HFR}$. MC4R protein expression (upper panel), cAMP (bottom left), and GLP-1 (bottom right) concentrations in the culture medium were determined by ELISA. (K) STC-1 cells were treated with saline, nesfatin-1 WT or Nesfatin-1$^{\Delta HFR}$. MC4R protein expression (upper panel), cAMP (bottom left), and GLP-1 (bottom right) concentrations in the culture medium were determined by ELISA. n.s no significance. Values are shown as the mean ± SEM ($n = 4$–6 rats or 3–4 independent experiments). two-way ANOVA followed by Bonferroni's test was used for (B), one-way ANOVA followed by uncorrected Fisher's LSD test was used for (D), and one-way ANOVA followed by Bonferroni's test was used for (C, E–I), and one-way ANOVA followed by Tukey's test was used for (J, K). Source data are available online for this figure.

sufficient to reduce HGP and improve liver insulin sensitivity in the pre-absorption state. This finding indicated that when targeting the duodenum, nesfatin-1 can remotely reverse IR and improve glucose metabolism. However, we found that in NUCB2-KO rats fed an HFD, the inhibitory effect of intestinal glucose or lipid infusion on HGP disappeared, indicating that under HFD conditions, NUCB2-KO impairs intestinal nutrient-sensing mechanisms and exacerbated hepatic IR. However, the mechanism needs further study.

On the basis of prior and current findings, our understanding of the role of gut hormones in regulating HGP through the gut-brain-liver neural circuit has advanced. Despite these advances, the precise mechanisms underlying the transmission of hormone signals within intestinal epithelial cells and initiation of nerve impulses to the CNS remain poorly understood. In the current study, with global NUCB2/nesfatin-1 knockout, we observed that intestinal nesfatin-1 signaling decreased HGP and increased hepatic insulin sensitivity through the gut-brain-liver neural circuit. Interestingly, the inhibitory effect of intestinal nesfatin-1 on HGP was eliminated upon co-administration of nesfatin-1 and Ex-9. This finding prompted us to consider the role of GLP-1, given that GLP-1R-positive neurons are very similar to intestinal GLP-1-secreting cells, and the nodose ganglia, where GLP-1R is expressed, house the vagal afferent cells innervating the intestine. Previously, our study established that GLP-1 prompts the activation of GLP-1 receptors in the gut, thus initiating the motion of the gut-brain-liver neural circuit. This subsequently leads to a decrease in HGP and an elevation in hepatic insulin sensitivity in vivo (Yang et al, 2017), suggesting the integral role of GLP-1 in intestinal function and the metabolic axis of the gut-brain-liver. Thus, we postulated that the regulation of HGP by intestinal nesfatin-1 may depend on GLP-1 or its receptor GLP-1R. However, a co-immunoprecipitation experiment revealed a lack of interaction between nesfatin-1 and GLP-1R. Nonetheless, recent studies reported the co-localization of nesfatin-1 and GLP-1 in immunopositive cells within the mouse intestine and revealed that nesfatin-1 appeared to stimulate GLP-1 release and secretion in enteroendocrine L cells (Ramesh et al, 2015; Ramesh et al, 2016). Further exploration of the relationship between nesfatin-1 and the GLP-1 signaling pathway was conducted via in vitro experiments. Our results revealed a significant increase in GLP-1 levels in STC-1 cells and the culture medium of intestinal tissue fragments following treatment with nesfatin-1. These findings suggest a potential connection between intestinal nesfatin-1 and GLP-1 production. However, Panaro et al reported that activation of the intestinal MC4R does not increase GLP-1 levels in rats (Panaro et al, 2014). We believe that the

difference between the results of Panaro et al and our results is mainly due to the use of different experimental methods: (1) Panaro et al used mesenteric artery infusion for drug administration, while we used local stimulation in the intestine; (2) Panaro et al used short-term infusion (8 min), while we stimulated for 24 h; (3) Panaro et al measured circulating GLP-1 levels, while we measured GLP-1 concentrations in the culture media of local tissues and cells; and (4) the methods for measuring GLP-1 levels are different. However, further in vitro and in vivo research is needed. Overall, we demonstrated a positive regulatory role for nesfatin-1 in GLP-1 secretion in STC-1 cells. Therefore, targeting nesfatin-1 to restore GLP-1 secretion may be a promising therapeutic strategy for T2D management.

In physiological and metabolic stress situations, well-established activation of cAMP signaling is triggered by the release of hormones, including adrenaline and glucagon, resulting in an increase in intracellular cAMP levels (Aslam and Ladilov, 2022). Notably, previous research has confirmed that intestinal cAMP signaling activates the secretion of GLP-1 (Li, Zhu et al, 2019). Our in vitro findings revealed an increase in cAMP levels concurrent with increased GLP-1 production following nesfatin-1 administration in both STC-1 cells and the culture medium of intestinal tissue fragments. Therefore, it is plausible that intestinal nesfatin-1 promotes increases in intracellular cAMP levels in intestinal L cells, and this cascade of events results in increased GLP-1 secretion, subsequently leading to the inhibition of HGP. Therefore, we hypothesized that cAMP-GLP-1 signaling is necessary for the regulation of HGP by intestinal nesfatin-1. However, a deeper understanding of the underlying molecular mechanisms requires further investigation.

The fundamental role of the melanocortin system in maintaining energy homeostasis within the CNS is well established, exerting profound effects on variables such as food intake, body weight, and energy expenditure (Ellacott and Cone, 2006; Jeong et al, 2014). NUCB2/nesfatin-1 has been found to engage with the melanocortin system within the CNS, which is integral to the regulation of food intake. Compelling evidence underlining its importance comes from rodent models in which intervention with the MC3/4R antagonist SHU9119, either intracerebroventricularly or into the lateral parabrachial nucleus, effectively negates the food intake reduction induced by the central injection of nesfatin-1 (Cheung et al, 2009; Yuan et al, 2017), underscoring the relationship. Research has revealed that the melanocortin pathway is also involved in the function of NUCB2/nesfatin-1 in the regulation of energy consumption and central hypertensive actions (Dore, Levata et al, 2017; Stephan et al, 2022; Yosten and Samson, 2014). A critical player in the melanocortin system is MC4R, a GPCR that orchestrates energy homeostasis

primarily within the CNS (Andermann and Lowell, 2017; Baldini and Phelan, 2019; Wei, Li et al, 2023). In addition to being expressed in the CNS, MC4R is also expressed in the enteroendocrine L cells of the intestine (Panaro et al, 2014), which can stimulate L cells to release GLP-1 to lower body weight and improve glucose metabolism (Holst, 2007; Panaro et al, 2014). Panaro et al reported that MC4R is mainly expressed in the basolateral side of enteroendocrine cells (Panaro et al, 2014). Therefore, how nesfatin-1 can reach the basolateral side in enteroendocrine cells and bind to MC4R is currently unclear. In an early report, Pusztai reported that small intestinal cells could transport macromolecules across cells through passive transport, endocytosis, or through paracellular pathways to the basolateral side of cells (Pusztai, 1989). However, its mechanism needs further study. Nonetheless, our findings reinforce the vital role of peripheral MC4R in the modulation of hepatic glucose metabolism. We found that SHU9119 infusion eliminated the effect of intestinal nesfatin-1 on GIR and HGP, suggesting a role for MC4R in intestinal nesfatin-1 function. Given that cAMP, a secondary messenger controlling various cellular processes, is downstream of MC4R (Molden et al, 2015; Yang and Harmon, 2017), we assessed the impact of nesfatin-1 on cAMP signaling. Our data reveal that nesfatin-1 treatment led to elevated cAMP levels, aligning with the role of MC4R. These findings indicate that the function of nesfatin-1 depends on the MC4R-cAMP-GLP-1 pathway for the regulation of HGP.

Regarding the interaction between NUCB2/nesfatin-1 and MC4R, although previous studies have suggested a potential link between NUCB2/nesfatin-1 and MC4R (Rojczyk et al, 2015; Ying et al, 2015), the significance of their interaction in the intestine and the exact binding domain has never been explored. This gap in our understanding informed our multifaceted approach to explore the potential interactions between NUCB2/nesfatin-1 and MC4R. Our bioinformatic analysis and IP-MS results provide preliminary indications of the potential interaction between NUCB2/nesfatin-1 and MC4R. Subsequent experimental observations strengthened this hypothesis: IF staining revealed the co-localization of NUCB2/nesfatin-1 and MC4R within the small intestinal tissue. Furthermore, to bolster these findings, co-IP and FRET experiments verified the direct interactions between these proteins in STC-1 cells. While we discerned an interaction between NUCB2/nesfatin-1 and MC4R, the specific regions implicated in this interaction requires further investigation. Previous research has suggested that the mid-segment of nesfatin-1 may regulate feeding and metabolism (Shimizu et al, 2009a), thus possibly serving as an active domain of nesfatin-1. Notably, our deletion mutation analysis in STC-1 cells identified the 51–75 aa region of NUCB2/nesfatin-1 as a key binding site for MC4R. This region contains an amino acid sequence (H-F-R; Phe-Arg-Trp) similar to those of α-, β-, and γ-MSH, which are endogenous agonists of MC4R. This sequence was proposed to possess the characteristic motif of the MC4R recognition site (Yang and Harmon, 2017). A mutation in this H-F-R sequence abrogated the NUCB2/nesfatin-1-MC4R interaction and reduced its inhibitory effect on HGP, underscoring the importance of this domain. Overall, our work establishes the H-F-R domain of NUCB2/nesfatin-1 as essential for its physiological function, providing, for the first time, the molecular mechanism underlying the metabolic regulation of NUCB2/nesfatin-1.

In conclusion, our study advances our knowledge of metabolic regulation by providing clear evidence that NUCB2/nesfatin-1

directly binds to MC4R in intestinal epithelial cells. This binding event initiates the cAMP signaling cascade, facilitating the release of GLP-1. The resulting NUCB2/nesfatin-1-MC4R-cAMP-GLP-1 signaling axis engages a gut-brain-liver neural network, which ultimately diminishes HGP in vivo. Our findings strengthen the understanding of the role of intestinal hormones in regulating glucose metabolism, clarify the signal transduction mechanism of hormones in the intestinal epithelium, and reveal a cellular and molecular signaling pathway to the intestine-brain-liver neural network for HGP regulation in rodents.

# Methods

**Reagents and tools table**

| Reagent/resource | Reference or source | Identifier or catalog number |
|---|---|---|
| **Experimental models** | | |
| NUCB2-KO rats (Sprague-Dawley background) | CYAGEN, Inc., China | ENSRNOG 00000020456 |
| **Recombinant DNA** | | |
| Plasmid sequence designed for co-IP experiment | Youbio Biological Technology Co., Ltd | Supplementary Table S2 |
| Plasmid sequence designed for mutagenesis | Youbio Biological Technology Co., Ltd | Supplementary Table S3 |
| ScAAV-U6-shRNA (Mc4r)-GMV-mCherry-tWPA | OBiO, Inc., Shanghai, China | Y32299 |
| ScAAV-*mCherry* | OBiO, Inc., Shanghai, China | M0508 |
| Ad-MC4R | OBiO, Inc., Shanghai, China | H07430 |
| Ad-shMC4R | OBiO, Inc., Shanghai, China | Y36589 |
| **Antibodies** | | |
| Anti-NUCB2/nesfatin-1 antibody | Abcam, USA | ab229683 |
| Anti-MC4R antibody | Abcam, USA | ab24233 |
| Goat anti-rabbit antibody | Abcam, USA | ab6721 |
| Anti-NUCB2/nesfatin-1 antibody | R&D Systems Biotechnology, USA | AF6895 |
| Anti-GLP-1 antibody | Abcam, USA | ab108443 |
| Anti-MC4R antibody | Abcam, USA | ab106596 |
| Donkey Anti-Sheep IgG H&L (FITC) antibody | Abcam, USA | ab6896 |
| Goat Anti-Mouse IgG H&L (FITC) antibody | Abcam, USA | ab6785 |
| Goat Anti-Rabbit IgG H&L (Alexa Fluor® 647) antibody | Abcam, USA | ab150079 |
| Anti-PEPCK antibody | Santa Cruz Inc., USA | sc27957 |
| Anti-G6Pase antibody | Abcam, USA | ab83690 |
| Anti-InsR antibody | Cell Signaling Technology | #3025 |
| Anti-phospho-InsR (Tyr 1150/ 1151) antibody | Cell Signaling Technology | #3024 |
| Anti-IRS-1 antibody | Cell Signaling Technology | #2382 |

| Reagent/resource | Reference or source | Identifier or catalog number |
|---|---|---|
| Anti-phospho-IRS-1 (Ser 302) antibody | Cell Signaling Technology | #2384 |
| Anti-Akt antibody | Cell Signaling Technology | #9272 |
| Anti-phospho-Akt (Ser 473) antibody | Cell Signaling Technology | #9271 |
| Anti-β-actin antibody | Santa Cruz Inc., USA | sc-58673 |
| Anti-MC4R antibody | Santa Cruz Inc., USA | sc-55567 |
| Anti-Flag antibody | Abcam, USA | ab1162 |
| Anti-HA antibody | Abcam, USA | ab137838 |
| **Oligonucleotides and other sequence-based reagents** | | |
| Primer sequences for quantitative RT-PCR | Accurate Biotechnology, China | Supplementary Table S1 |
| **Chemicals, enzymes and other reagents** | | |
| Normal chow diet | New Brunswick, NJ, USA | D1245B |
| High-fat diet | New Brunswick, NJ, USA | D1245 |
| Nesfatin-1 protein | USCN Business Co. Ltd., Wuhan, China | #APA242Ra01 |
| SHU9119 | Sigma-Aldrich Inc., MO, USA | #M4603 |
| Tetracaine | Sigma–Aldrich Inc., MO, USA | PHR3140 |
| Exendin fragment 9–39 | Aladdin, Shanghai, China | E421270 |
| MK-801 | Sigma–Aldrich Inc., MO, USA | 475878 |
| Triton 100 | Sigma–Aldrich Inc., MO, USA | 93443 |
| DAPI | Abcam, USA | ab104139 |
| GLP-1 ELISA kit | Millipore Corporation, Billerica, USA | #EZGLP1T-36K |
| cAMP ELISA kit | BioVision Inc., USA | #K371-100 |
| TRIzol reagent | Accurate Biotechnology, China | AG21102 |
| Protein A/G magnetic beads | Med Chem Express, China | HY-K0202 |
| **Software** | | |
| ImageJ software | National Institutes of Health, USA | V1.8.0 |
| QE1 system | Shanghai Science Research Center | 1.0 |
| Graphpad | GRAPHPAD SOFTWARE, LLC,USA | 9.0 |
| **Other** | | |

## Animal preparation

NUCB2-KO rats (Sprague-Dawley background) were generated using the CRISPR/Cas9 technique (CYAGEN, Inc., China). Briefly, 14 exons were identified, with the ATG start codon in exon 3 and the TAA stop codon in exon 14 (Transcript: ENSRNOG 00000020456). The sequences of exons 7–8 were selected as target sites. Cas9 mRNA and gRNA were generated by in vitro transcription and injected into fertilized eggs to produce NUCB2-KO rats (Fig. S8A). The founders were genotyped using PCR to confirm accurate knockout (Fig. S8B). Male SD rats aged 8 weeks were purchased from the Animal Center of Chongqing Medical University. The animals were kept in a temperature- and humidity-controlled facility with a 12 h light/dark cycle and free access to water and food. Male SD and NUCB2-KO rats were fed an NCD (D1245B) or an HFD (60% fat; D1245, Research Diets, New Brunswick, NJ, USA) for 12 weeks. Rats were randomly assigned to various experimental groups and no blinding was done. All experimental procedures were approved by the Animal Experimental Ethics Committee of Chongqing Medical University (Approval No. 2016-32).

## Catheterization of the duodenum, jugular vein, and carotid artery

Four days prior to the insulin clamp study, a cannula was inserted into the duodenum ~2 cm downstream of the pyloric sphincter in experimental rats, as previously reported (Lin et al, 2019). For the PEC study, duodenal cannulation and internal jugular vein and carotid artery catheterizations were performed for infusion and blood sample collection, as previously reported (Yang et al, 2017).

## Stereotaxic surgery and HBV

For stereotactic surgery, a catheter was implanted into the NTS in rats one week prior to duodenal and vascular cannulation. The location was 0.0 mm from the occipital apex, 0.8 mm lateral to the midline, and 7.9 mm below the craniofacial region (Wu et al, 2014). HBV or sham operation (SHAM) was performed in the two subgroups of NUCB2-KO rats simultaneously for duodenal catheter insertion, as previously described (Yang et al, 2017).

## PEC procedure

Four days after internal jugular vein and carotid artery catheterization, PEC was performed in conscious and unrestrained rats. [3-H$^3$] Glucose (0.2 µCi/min) infusion started at 0 min and continued until the end of PEC. To inhibit endogenous insulin secretion, somatostatin (3 µg/kg/min), insulin (1.2 mU/kg/min), and 25% glucose were infused intravenously at the beginning of the PEC. The blood glucose level was adjusted every 5 min to maintain it at ~6 mmol/L (Zhang, Luo et al, 2019).

## Gut and other treatments

To investigate the dose-dependent effects of gut nesfatin-1 on GIR, different concentrations of nesfatin-1 protein (0, 0.5, 1, 2, 4 µg/kg/min) (Ca: #APA242Ra01, USCN Business Co. Ltd., Wuhan, China) or nesfatin-1$^{ΔHFR}$ (2 µg/kg/min) (customized by USCN Business Co. Ltd., Wuhan, China) were infused into the duodenum. In a preliminary experiment, when the infusion rate of nesfatin-1 was 100 µg/kg, the GIR peaked. Therefore, in subsequent experiments, we used 100 µg/kg nesfatin-1 (2 µg/kg/min) for the duodenal infusion (Fig. S2G). For the MC4R inhibition study, different concentrations of SHU9119 (0, 10,

30, 60, and 90 nmol/kg/min) (Ca: #M4603, Sigma-Aldrich Inc., MO, USA) were infused into the intestine at the same time as nesfatin-1 (Fig. S9A). The GIR value was the lowest when the infusion rate of SHU9119 was 60 nmol/kg/min. The GIR did not decrease further in response to 90 nmol/kg/min SHU9119 (Fig. S9B). Furthermore, the following materials were infused into the gut during PEC: (1) saline; (2) glucose (4 nmol/min, 2 mM; Peace Pharmaceutical Co., Ltd., China) (Bauer et al, 2018); (3) 20% intralipids (0.03 kcal/min, Sino-Swed Pharmaceutical Corp. Ltd, Beijing, China); (4) tetracaine (0.01 mg/min); and (5) the GLP-1 receptor antagonist exendin fragment 9–39 (Ex-9) (300 pmol/kg/min, Aladdin, Shanghai, China). For the NTS experiments, MK-801, an NMDA receptor inhibitor (0.03 ng/min, Sigma‒Aldrich Inc., MO, USA), was continuously infused into the NTS from $t = 90$ min until the end of the PEC.

## Fasting-feeding tests

The rats were fasted for 40 h beginning at 5:00 p.m. before the start of the experiment. The rats were then infused with saline or nesfatin-1 (2 µg/kg/min) into the duodenum for 30 min and fed normally for 20 min after fasting (Fig. S3A). Blood glucose levels were measured at the indicated time points (0, 10, and 20 min) (Fig. S3B). Food intake was measured for 20 min at the end of the study (Fig. S3C).

## Analytical procedures

The glucose oxidase method and ELISA were used to measure blood glucose and insulin levels. FFA levels were determined using a commercial kit (Randox Laboratories, Ltd., Antrim, UK). The serum TG and TC levels were determined using an enzymatic colorimetric kit. The specific activity of [3-$^3$H] glucose in the serum was determined using a scintillation counter, as previously reported (Wu et al, 2014).

## IHC and IF staining

For IHC analysis, duodenal sections were pretreated with 0.5% Triton 100 (Sigma-Aldrich) and 1.5% BSA for 15 min at 25 °C. The sections were subsequently incubated with anti-NUCB2/nesfatin-1 (ab229683; Abcam) and anti-MC4R (ab24233; Abcam) polyclonal antibodies. Immunoreactivity was detected using horseradish peroxidase-labeled goat anti-rabbit antibody (ab6721; Abcam).

For IF analysis, frozen sections of gut tissue were incubated with primary antibodies, including anti-NUCB2/nesfatin-1 (diluted 1:500; AF6895; R&D Systems Biotechnology, USA), anti-GLP-1 (diluted 1:100; ab108443; Abcam, USA) or anti-MC4R (diluted 1:50; ab106596; Abcam), at 4 °C overnight and then washed with PBS. The sections were incubated with appropriate secondary antibodies (diluted 1:1000; ab6896, ab6785 or ab150079, Abcam) and counterstained with DAPI (ab104139, Abcam), as previously described (Zhang et al, 2019). Images of fluorescence-labeled gut sections were obtained with a Nikon A1R confocal microscope. Images were captured, and fluorescence intensities were calculated via ImageJ software.

## PPI network construction and KEGG analysis

To construct the protein–protein interaction (PPI) network of NUCB2/nesfatin-1, we utilized the STRING database (version 11.1; https://string-db.org), selecting first-degree interactors with an interaction score > 0.04 as per STRING guidelines. KEGG enrichment analysis was performed using the STRING database.

## scAAV infection

For the cAMP and GLP-1 secretion experiments in vivo, NUCB2-KO rats were anesthetized with ketamine (60 mg/kg), and surgeries were performed. A 6 cm segment of the upper small intestine was washed three times with physiological saline and sutured at both ends. Then, scAAV-U6-shRNA (Mc4r)-GMV-mCherry-tWPA (scAAV-shMC4R) or control scAAV-mCherry ($1 \times 10^{11}$ viral particles in 200 µl of saline per rat, OBiO, Inc., Shanghai, China) was injected into the intestinal wall. The sutures were removed after a 20 min incubation. Ten days post-transfection, the small intestine was taken for subsequent experiments.

## Cell and ex vivo intestinal tissue culture and treatment

Mouse enteroendocrine STC-1 cells, which exhibit a phenotype similar to that of enteroendocrine L cells, and HEK293T cells were cultured in high-glucose Dulbecco's modified Eagle's medium (DMEM) supplemented with 10% fetal bovine serum (FBS) and penicillin-streptomycin solution in a humidified atmosphere of 5% $CO_2$ at 37 °C and passaged every 2 days by trypsinization. For the in vitro GLP-1 and cAMP secretion experiments, small intestinal fragments from NUCB2-KO rats were isolated and incubated in DMEM supplemented with 10% FBS and 1% penicillin-streptomycin. To determine the optimal concentrations of nesfatin-1 and/or SHU9119, cultured STC-1 cells or small intestinal fragments were treated with nesfatin-1 (0, 0.01, 0.1, and 1 µM) or nesfatin-1 (0.1 µM) plus SHU9119 (0, 1, 10 and 100 µg/ml) or nesfatin-1$^{\Delta HFR}$ (0.1 µM) for 1 h or 24 h, respectively. For cAMP and GLP-1 secretion experiments in vitro, STC-1 cells were transfected with Ad-MC4R/ shMC4R ($10^7$ PFU/mL) for 24 h. The virus sequence was as follows: F: TACGATACAAGGCTGTTA-GAGAG; R: CTATTAATAACTAATGCATGGC for Ad-MC4R; Target Seq: CCATCGTCATTACCCTGTTAA for Ad-shMC4R (Obio Technology Co. Ltd., Shanghai, China). Ad-mCherry was used as a control. The GLP-1 or cAMP concentrations were determined using an ELISA kit (Catalog # EZGLP1T-36K; Millipore Corporation, Billerica, USA; #K371-100; BioVision Inc., USA) according to the manufacturer's instructions, as described in a previous report (Lai et al, 2020).

## Quantitative RT–PCR

Total RNA was extracted using TRIzol reagent (AG21102, Accurate Biotechnology, China) according to the manufacturer's instructions. qRT-PCR was performed as previously reported (Lai et al, 2020). The primer sequences are shown in Table S1.

## Western blotting

Protein expression was measured by western blotting as described previously (Sharma et al, 2020). The primary antibodies used were anti-PEPCK (sc27957, Santa Cruz Inc., USA), anti-G6Pase (ab83690, Abcam), anti-InsR/phospho-InsR (Tyr 1150/1151) (#3025, #3024, Cell Signaling Technology), anti-IRS-1/phospho-

IRS-1 (Ser 302) (#2382, #2384, Cell Signaling Technology), anti-Akt/phospho-Akt (Ser 473) (#9272, #9271, Cell Signaling Technology), anti-NUCB2/nesfatin-1 (ab229683, Abcam), anti-MC4R (ab24233, Abcam), and β-actin (Santa Cruz Inc., USA).

## Construction of plasmid vectors

Plasmids expressing NUCB2 (pc-NUCB2-Flag), MC4R (pc-MC4R-HA), GLP-1R (pc-GLP-1R-HA), or empty vectors were purchased from Gene Chem Co. (Shanghai, China). The plasmid sequences are listed in Table S2. The plasmids pc-ΔEF-Flag (NUCB2:247aa-322 aa mutant), pc-ΔDNA-Flag (NUCB2:171aa–223 aa mutant), pc-ΔN25–50-Flag (NUCB2:25aa–50 aa mutant), pc-ΔN51–75- Flag (NUCB2:51aa–75 aa mutant), pc-ΔN76–106-Flag (NUCB2:76aa–106 aa mutant), and pc-NUCB2$^{\Delta HFR}$-Flag (NUCB2-$_{70}$H-F-R$_{72}$ mutant) were purchased from YouBio Biological Technology Co., Ltd. (Hunan, China). The plasmid sequences are listed in Table S3. All plasmids were confirmed by DNA sequencing (Zhou et al, 2020).

## Immunoblotting and co-IP experiments

For co-IP experiments, STC-1 cells were transfected or cotransfected with pc-NUCB2-Flag pc-MC4R-HA, pc-GLP-1R-HA, or empty vector for 48 h. For binding site analysis, HEK293T cells were transfected or cotransfected with pc-MC4R-HA, pc-NUCB2-Flag, pc-ΔEF-Flag, pc-ΔDNA-Flag, pc-Δ25-50-Flag, pc-Δ51–75-Flag, pc-Δ76-106-Flag, pc-ΔHFR-Flag, or empty vector for 48 h. Then, the cells were lysed with RIPA buffer containing protease/phosphatase inhibitors. The cell lysates were incubated with the indicated antibodies, including anti-NUCB2/nesfatin-1 (ab229683; Abcam), anti-MC4R (sc-55567; Santa Cruz Biotechnology, Inc.), anti-Flag (ab1162; Abcam), anti-HA antibodies (ab137838; Abcam), and protein A/G magnetic beads (HY-K0202; Med Chem Express). The immunoprecipitated complexes or cell lysates were separated by SDS-PAGE, transferred to nitrocellulose membranes, and subjected to immunoblotting with the indicated antibodies, as described previously (Yu et al, 2019; Zhao et al, 2022).

## IP-MS analysis

Pc-NUCB2-Flag or an empty vector was transfected into STC-1 cells. The cells were harvested 48 h after transfection, and the cell lysates were immunoprecipitated with an anti-Flag antibody. After elution, the samples were separated by SDS-PAGE and stained with Coomassie Brilliant Blue. NUCB2 and empty vector bands were cut and analyzed by MS using the QE1 system (Shanghai Science Research Center). Data of MS are freely available by contacting authors.

## FRET

The FRET experiment refers to the phenomenon that when the distance between these two fluorescent groups was less than 10 nm, the energy between the groups was transferred from the donor to the receptor in a nonradiative manner through dipole coupling (Fig. 6G). EGFP-mCherry fluorescent group pairs were selected for the experiment using the receptor photobleaching method (Yu et al, 2019). The negative controls were pEGFP-N1 and pmCherry-N1, and the probability of energy transfer between them was very small. The experimental plasmids used were pEGFP-NUCB2 and pmCherry-MC4R (Fig. 6H). The above plasmids were transfected into HEK293T cells for 48 h. Then, the cells were seeded and cultured in a 35 mm confocal microscopy dish and fixed with 4% paraformaldehyde for 30 min. Experimental plasmid information is provided in Table S2. Images were captured in both channels before and after the photo was bleached. In all experiments, approximately 5–6 cells were measured, and FRET efficiency was calculated as E = (1-pre/post) × 100%, where Pre and Post represent the fluorescence intensities of the donor (GFP) before and after photobleaching, respectively. Images were obtained using a confocal laser microscope (Leica TCS SP8, Germany) with the LAS ×FRET AB module.

## MD studies

Structural models of MC4R and NUCB2 were obtained from the Protein Data Bank (www.rcsb.org, ID:6w25) and SWISS-MODEL online tools (ID: P80303), respectively (Waterhouse et al, 2018; Yu et al, 2020; Zhao et al, 2022). NUCB2 (51–75 aa, NUCB2-$_{70}$H-F-R$_{72}$) was manually cut to construct a complex with MC4R using Discovery Studio 3.5 according to the cocrystal ligand. The $_{70}$H-F-R$_{72}$ region of NUCB2 was mutated to $_{70}$A-A-A$_{72}$, named NUCB2_ AAA, and constructed using PyMOL as a negative control. The two complexes were dissolved in 0.15 mol/L NaCl, and a 100 ns MD simulation was performed in a cubic box with periodic boundary conditions using the AMBER16 software package (Genheden and Ryde, 2015; Maier et al, 2015). For the stable MC4R/NUCB2 conformation achieved by MD simulation, the last 200 snapshots were used for binding free energy ($\Delta G_{bind}$) analysis using the MM/GBSA method as follows: $\Delta G_{bind}$ = $\Delta H$-T$\Delta S$ ≈ $\Delta E_{MM}$ + $\Delta G_{SOI}$ − T$\Delta S$. $\Delta E_{MM}$, gas-phase MM energy during binding; $\Delta G_{SOI}$, change in solvation-free energy; T$\Delta S$, the entropy of the molecule (Hu et al, 2016; Ryckaert et al, 1977).

## Statistical analysis

All experiments subjected to statistical tests were conducted a minimum of three times as biological replicates. No data were excluded from the analyses. Statistical analyses were performed using Prism software version 9.0 (GraphPad). Comparisons between the two groups were performed using unpaired Student's $t$-tests. One-, two-, or three-way ANOVA followed by Bonferroni's test, Tukey's test, or Uncorrected Fisher's LSD test was used for comparisons among multiple groups. All the results are expressed as the means ± SEM. Statistical significance was set at $p < 0.05$. The data between T = 60–90 min and T = 180–200 min under basal and clamp conditions were averaged.

# Data availability

All data generated in this study are available within the article, supplementary information, or source data file. Source data are provided with this paper. No data amenable to large-scale repository deposition are included in this study.

The source data of this paper are collected in the following database record: biostudies:S-SCDT-10_1038-S44318-024-00300-4.

# Peer review information

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

## Acknowledgements

This study was supported by grants from the National Natural Science Foundation of China (no. 81670755, 82270853, 82370852 and U22A20289). Postdoctoral Science Foundation of China (No. 2020M683263) and Natural Science Foundation Project of Chongqing, Chongqing Science and Technology Commission (cstc2021jcyj-msxmX0353). The Scientific and Technological Research Program of Chongqing Municipal Education Commission (KJZD-K202200401) and the CQMU Program for Youth Innovation in Future Medicine (W0159).

## Author contributions

**Shan Geng**: Data curation; Software; Formal analysis; Funding acquisition; Investigation; Visualization; Methodology. **Shan Yang**: Data curation; Software; Formal analysis; Validation; Investigation; Visualization; Methodology. **Xuejiao Tang**: Data curation; Software. **Shiyao Xue**: Data curation; Visualization. **Ke Li**: Formal analysis; Validation. **Dongfang Liu**: Formal analysis; Validation. **Chen Chen**: Writing—review and editing. **Zhiming Zhu**: Writing—review and editing. **Hongting Zheng**: Writing—review and editing. **Yuanqiang Wang**: Software; Methodology. **Gangyi Yang**: Resources; Supervision; Funding acquisition; Writing—original draft; Project administration; Writing—review and editing. **Ling Li**: Resources; Supervision; Funding acquisition; Project administration; Writing—review and editing. **Mengliu Yang**: Conceptualization; Resources; Supervision; Funding acquisition; Writing—original draft; Project administration; Writing—review and editing.

Source data underlying figure panels in this paper may have individual authorship assigned. Where available, figure panel/source data authorship is listed in the following database record: biostudies:S-SCDT-10_1038-S44318-024-00300-4.

## Disclosure and competing interests statement

The authors declare no competing interests.

