## [Peer Review File · The EMBO Journal]

Intestinal NUCB2/nesfatin-1 regulates hepatic glucose production via the MC4R-cAMP-GLP-1 pathway

Mengliu Yang, Shan Geng, Shan Yang, Xuejiao Tang, Shiyao Xue, Ke Li, Dongfang Liu, Chen Chen, Zhiming Zhu, Hongting Zheng, Yuanqiang Wang, Gangyi Yang, and Ling Li

Corresponding authors: Mengliu Yang (mengliu.yang@cqmu.edu.cn), Ling Li (liling@cqmu.edu.cn), Gangyi Yang (gangyiyang@hospital.cqmu.edu.cn)

Review Timeline:

Submission Date:	3rd Nov 23
Editorial Decision:	20th Feb 24
Revision Received:	15th Aug 24
Editorial Decision:	26th Sep 24
Revision Received:	14th Oct 24
Accepted:	23rd Oct 24

Editor: Daniel Klimmeck

Transaction Report:

Dear Dr Yang,

Thank you again for the submission of your manuscript (EMBOJ-2023-116057) to The EMBO Journal. As mentioned earlier, your study was assessed by two reviewers with expertise in systemic nutrient metabolism, endocrinology, and gastrointestinal signaling, whose comments are enclosed below.

As you will see from the experts' reports, the referees acknowledge the analysis and potential interest and value of your findings. However, they also express major concerns regarding clarity and robustness of the findings, which need to be addressed thoroughly to make them supportive of publication in the EMBO Journal. The reviewers also raise a number of issues related to the presentation of the findings, additional controls and improved methods annotation required, statistics applied and overall discussion of related literature, that would need to be conclusively addressed to achieve the level of robustness and clarity needed for The EMBO Journal.

Given the overall interest stated and broader angle of your findings, we are able to invite you to revise your manuscript experimentally to address the referees' comments. I need to stress though that we do require strong support from the referees on a revised version of the study in order to move on to publication of the work.

In light of the extensive experimentation requested, I would appreciate if you could contact me during the next weeks for exchange e.g. a video call to discuss your perspective on the comments and potential plan for revisions.

Please feel free to contact me if you have any questions or need further input on the referee comments.

When submitting your revised manuscript, please carefully review the instructions below.

Please feel free to approach me any time should you have additional questions related to this.

Thank you for the opportunity to consider your work for publication.

I look forward to your revision.

Kind regards,

Daniel Klimmeck

Daniel Klimmeck, PhD
Senior Editor
The EMBO Journal

Instruction for the preparation of your revised manuscript:

- 1) a .docx formatted version of the manuscript text (including legends for main figures, EV figures and tables). Please make sure that the changes are highlighted to be clearly visible.
- 2) individual production quality figure files as .eps, .tif, .jpg (one file per figure).
- 3) a .docx formatted letter INCLUDING the reviewers' reports and your detailed point-by-point response to their comments. As part of the EMBO Press transparent editorial process, the point-by-point response is part of the Review Process File (RPF), which will be published alongside your paper.
- 4) a complete author checklist, which you can download from our author guidelines ([https://wol-prod-cdn.literatumonline.com/pb-assets/embo-site/Author Checklist%20-%20EMBO%20J-1561436015657.xlsx](https://wol-prod-cdn.literatumonline.com/pb-assets/embo-site/Author%20Checklist%20-%20EMBO%20J-1561436015657.xlsx)). Please insert information in the checklist that is also reflected in the manuscript. The completed author checklist will also be part of the RPF.

6) It is mandatory to include a 'Data Availability' section after the Materials and Methods. Before submitting your revision, primary datasets produced in this study need to be deposited in an appropriate public database, and the accession numbers and database listed under 'Data Availability'. Please remember to provide a reviewer password if the datasets are not yet public (see <https://www.embopress.org/page/journal/14602075/authorguide#datadeposition>).

7) Our journal encourages inclusion of *data citations in the reference list* to directly cite datasets that were re-used and obtained from public databases. Data citations in the article text are distinct from normal bibliographical citations and should directly link to the database records from which the data can be accessed. In the main text, data citations are formatted as follows: "Data ref: Smith et al, 2001" or "Data ref: NCBI Sequence Read Archive PRJNA342805, 2017". In the Reference list, data citations must be labeled with "[DATASET]". A data reference must provide the database name, accession number/identifiers and a resolvable link to the landing page from which the data can be accessed at the end of the reference. Further instructions are available at .

8) At EMBO Press we ask authors to provide source data for the main and EV figures. Our source data coordinator will contact you to discuss which figure panels we would need source data for and will also provide you with helpful tips on how to upload and organize the files.

Numerical data can be provided as individual .xls or .csv files (including a tab describing the data). For 'blots' or microscopy, uncropped images should be submitted (using a zip archive or a single pdf per main figure if multiple images need to be supplied for one panel). Additional information on source data and instruction on how to label the files are available at .

9) We replaced Supplementary Information with Expanded View (EV) Figures and Tables that are collapsible/expandable online (see examples in <https://www.embopress.org/doi/10.15252/embj.201695874>). A maximum of 5 EV Figures can be typeset. EV Figures should be cited as 'Figure EV1, Figure EV2" etc. in the text and their respective legends should be included in the main text after the legends of regular figures.

11) For data quantification: please specify the name of the statistical test used to generate error bars and P values, the number (n) of independent experiments (specify technical or biological replicates) underlying each data point and the test used to calculate p-values in each figure legend. The figure legends should contain a basic description of n, P and the test applied. Graphs must include a description of the bars and the error bars (s.d., s.e.m.).

We realize that it is difficult to revise to a specific deadline. In the interest of protecting the conceptual advance provided by the work, we recommend a revision within 3 months (20th May 2024). Please discuss the revision progress ahead of this time with the editor if you require more time to complete the revisions.

Referee #1:

This work reports that gut nesfatin-1 can sense gut nutrients and reduces hepatic glucose production (HGP). Mechanistically, NUCB2/nesfatin-1 interacted directly with melanocortin 4 receptor (MC4R) through its H-F-R domain and increased cAMP levels and glucagon-like peptide 1 (GLP-1) secretion in the intestinal epithelium, thus inhibiting HGP. The study is timely, and results are novel and interesting. The authors have used a large set of methodological tools to address an original hypothesis.

General comments.

1. Figs 1 and 2 show the pivotal role of intestinal nesfatin-1 in regulating gut nutrient sensing and modulating HGP under both standard and IR conditions. It seems surprising that HGP shows similar levels in rats fed a chow diet and HFD. How do the authors explain this?
2. The role of GLP1 as a mediator of the actions of nesfatin is very interesting and the MC4R-dependent action nicely fits with a previous study concerning the role of this receptor in the secretion of GLP1. However, there are other studies showing that direct stimulation of small intestinal Melanocortin-4-Receptors in mice and rats does not affect GLP-1 secretion (PMID: 34421821). Thus, I think it is important to strengthen this part of the manuscript:
 - a) Since the authors have found that nesfatin increases GLP1 while SHU9119 reduces GLP1, I would recommend measuring GLP1 in Figs 4e and 4f, which only focused on cAMP.
 - b) I would recommend overexpressing MC4R and corroborate that GLP1 is augmented.
 - c) The functional interaction between MC4R and NUCB2 has been established by using the pharmacological inhibitor SHU, which inhibits both MC3R and MC4R. I would recommend using a more specific approach inhibiting specifically MC4R (i.e. virogenetic tools, CRISPR) in the duodenum.
 - d) Likewise, in vitro experiments should be done using more specific tools for M4R and see whether the action of nesfatin on nGLP1 levels is still reduced.
3. Fig 7i is not convincing. The photo clearly shows that protein levels of MC4R are increased in mutant Nesfatin-1-treated mice compared to saline, but the figure represents identical values in these 2 groups. Also, these results should be corroborated in vitro by measuring MC4R in intestinal fragments and STC1 cells incubated with WT and mutant nesfatin.
4. Since the study is focused in the effect of nesfatin-MC4R-GLP1 in intestinal cells, perhaps it would be relevant to also assess intestinal gluconeogenesis, which is activated by nutrients capable of fuelling systemic gluconeogenesis.
5. The last part addressing the role of duodenal nesfatin-1 on HGP via the gut-brain-liver neural circuit is also relevant. However, in this section MC4R and GLP1 are totally missed. It would be important to measure both after the different treatments/surgery.

Referee #2:

These studies by Geng and colleagues propose a role for intestinal nesfatin-1 as a gut nutrient sensor that regulates hepatic glucose production (HGP) via a novel gut-brain-liver axis. Using duodenal infusions, pancreatic clamps, and a novel nesfatin-1 knockout rat line, the authors build a case for intestinal nutrient entry increasing gut nesfatin-1 levels, and this subsequently reduces HGP in a glucagon-like peptide-1 (GLP-1) dependent manner. Additional experiments suggest that nesfatin-1 mediates this effect by interacting with the melanocortin-4 receptor (MC4R). Furthermore, evidence is presented suggesting that gut nesfatin-1 action regulates HGP by engaging the central nervous system. Taken together, these studies propose a unique model for a gut hormone to act as a nutrient sensor via a novel inter-organ mechanism. Although these studies are novel, there are significant issues that should be addressed to more precisely interpret the results:

Major comments:

1. These studies rely extensively on a newly created nesfatin-1/NUCB2 knockout rat model. Additional details and data

validating this model are needed. The only data shown for the generation of this model (Figure S6) only shows genomic DNA PCR results, and it is not quite clear what these results actually show. For example, PCR results from WT rats show a band at ~1.5kb and another at ~750kb. Are these bands expected, and why would the band at ~750kb that is present in WT rats also be present in the KO rats? Additional information along with showing that nesfatin-1 protein levels are reduced in various tissues in the KO rats is required.

2. Figure 1b shows that a 30 min glucose infusion increases gut nesfatin-1 levels. This appears to be too short of an infusion for it to be a transcriptional event. The authors should discuss potential mechanisms for increased nesfatin-1 protein expression under such a relatively short time period. Furthermore, the authors should include information about the rate of duodenal glucose infusion (the methods only state that 25% glucose was infused, but the total mass of glucose infused should be provided). Also, the results in Figure 1c are presented as "Protein expression (%)", but it is not clear what this means (i.e., % of what?).

3. For all clamp experiments, the entire time course of GIR should be presented. This is particularly important for assessing whether/to what degree the GIR was affected by the various duodenal infusions (glucose, lipid, nesfatin-1). As presented, only the last 20 minutes of GIR measurements during the 50 minute duodenal infusions are shown. The GIR throughout the entire 110 min clamp for all PEC experiments should be shown.

4. Regarding the PEC clamp experiments, the authors should discuss why HGP was suppressed even though euglycemia was maintained and insulin levels did not increase from the basal to the clamp period. This is more relevant in the high fat diet experiments in which insulin levels actually dropped during the clamp experiments yet HGP was still somewhat suppressed in all groups.

5. Results from high fat diet fed rats require additional discussion because they suggest that the loss of nesfatin-1 does not just prevent the suppression of HGP by duodenal glucose but it actually impairs hepatic insulin resistance even further. This should be addressed in the Discussion.

6. The duodenal nesfatin-1 infusion protocol requires additional information and validation. The authors state "To prevent an increase in circulating nesfatin-1 levels, a nesfatin-1 protein concentration of 100 ug/kg was selected for all subsequent experiments" (Page 7). One issue with this is that the 100 ug/kg is a concentration and not a rate of infusion. The rate of infusion should be provided. The other issue is that the authors should show whether this infusion does not increase circulating nesfatin-1 levels. This is an important result because it would validate the model that nesfatin-1 is acting locally in the gut.

7. There are several experiments that were conducted in NUCB2 KO rats but not in WT rats. This is a necessary control. For example, Fig. S2b shows that nesfatin-1 lowers glucose levels during a refeed in fast-refed NUCB2 KO rats. However, it is important to show: a) what are the glucose levels during the refeed period in a WT rat (are they comparable to those of the nesfatin-infused NUCB2 KO) and b) does nesfatin-1 further lower glucose levels during a refeed in WT rats. Similar controls in WT rats should be done for the GTTs, ITTs.

8. Panaro et al. Cell Metab 2014 (reference 22) show that the MC4R is expressed in the basolateral side in gut enteroendocrine cells. This would make it difficult for a duodenal infusion of nesfatin-1 to interact with gut MC4R unless there is a mechanism for duodenally-infused nesfatin-1 to be transported from the luminal side to the basolateral side intact. Furthermore, the same study showed that activation of the intestinal MC4R does not increase GLP-1 levels in rats and only does so in mice. These discrepancies should be discussed.

9. Fig. 5 shows increased nesfatin-1 and MC4R expression in duodenal epithelium following duodenal nesfatin-1 infusion. The assumption is that the increased nesfatin-1 is from the exogenous infusion since these experiments were conducted in NUCB2 KO rats. The authors should clarify this (this also partially addresses point #7 above). Furthermore, the authors should also provide data obtained from WT rats. This will show whether NUCB2 KO rats have lower baseline gut MC4R expression.

10. The experimental interventions used to test the gut-brain-liver connection should include validations. Evidence that tetracaine inhibited neurotransmission and MK-801 inhibited NMDA signaling should be provided. Also, the authors should specify whether these experiments were conducted in WT or NUCB2KO rats.

11. In general, all figure legends should describe the statistical test(s) used for the specific comparisons shown in that figure.

Minor comments:

1. In Page 5, the authors state "Throughout the PEC period, blood glucose, insulin, free fatty acid (FFA), and triglyceride (TG) levels remained stable at basal levels (Fig. S1a-e)" when referring to results obtained from experiments using duodenal saline/glucose infusions. However, the legend for Fig. S1a-e indicates that these are values obtained from experiments in which duodenal nesfatin-1 was infused.

2. In Page 5 the authors also state "In contrast, NUCB2 KO rats demonstrated a substantial reduction in GIR (Fig. 1e and f) and a corresponding increase in HGP (Fig. 1g and h)". It would be more accurate to state "a corresponding impairment in the

suppression of HGP". A similar modification should be made in Page 6, in the sentence "Conversely, lipid-infused NUCB2 KO rats demonstrated a notable decrease in GIR (Fig. 1k and l) and a significant increase in HGP (Fig. 1 m-n)".

3. There appears to be an incorrect statement in Page 7, "We hypothesized that duodenal nesfatin-1 may interfere with nutrient-sensing-related mechanisms activated by refeeding, thereby further inhibiting HGP and reducing blood glucose." Wouldn't the hypothesis be that nesfatin-1 "contributes to" and not interferes with nutrient-sensing-related mechanisms?

Response Letter (EMBOJ-2023-116057)

Dear Dr. Klimmeck,

We are grateful for these constructive comments from you and the reviewers on our manuscript entitled “Intestinal NUCB2/nesfatin-1 regulates hepatic glucose production *via* the MC4R-cAMP-GLP-1 pathway and gut-brain neural circuit”. These comments enriched the manuscript (MS) content and greatly improved its quality. We have revised the MS, changed and added figures. All changes are highlighted in red. For details, please find our point-by-point responses below.

Reviewer #1

This work reports that gut nesfatin-1 can sense gut nutrients and reduces hepatic glucose production (HGP). Mechanistically, NUCB2/nesfatin-1 interacted directly with melanocortin 4 receptor (MC4R) through its H-F-R domain and increased cAMP levels and glucagon-like peptide 1 (GLP-1) secretion in the intestinal epithelium, thus inhibiting HGP. The study is timely, and results are novel and interesting. The authors have used a large set of methodological tools to address an original hypothesis.

General comments.

- 1. Figs 1 and 2 show the pivotal role of intestinal nesfatin-1 in regulating gut nutrient sensing and modulating HGP under both standard and IR conditions. It seems surprising that HGP shows similar levels in rats fed a chow diet and HFD. How do the authors explain this?*

Response: We appreciate your careful review and fully understand your concerns regarding HGP. As reported previously [1-6], we established an obese insulin resistance model. Theoretically, although tissue insulin sensitivity is reduced in these rat, its hyperinsulinemia still compensatorily inhibits HGP. Thus, basal HGP was not high, and some studies even reported a slight decrease; however, generally, the difference was not statistically significant.

References

[1] White AT, LaBarge SA, McCurdy CE, Schenk S. Knockout of STAT3 in skeletal muscle

does not prevent high-fat diet-induced insulin resistance. *Mol Metab* 2015; 4: 569-575 (See **Fig. 3D**).

[2]Lee E, Jung DY, Kim JH, Patel PR, Hu X, Lee Y, Azuma Y, et al. Transient receptor potential vanilloid type-1 channel regulates diet-induced obesity, insulin resistance, and leptin resistance. *FASEB J* 2015; 29: 3182-3192 (See **Fig. 4E**).

[3]Lackey DE, Lazaro RG, Li P, Johnson A, Hernandez-Carretero A, Weber N, Vorobyova I, et al. The role of dietary fat in obesity-induced insulin resistance. *Am J Physiol Endocrinol Metab* 2016; 311: E989-E997 (See **Fig. 3C**).

[4]Chao LC, Wroblewski K, Zhang Z, Pei L, Vergnes L, Ilkayeva OR, Ding SY, et al. Insulin resistance and altered systemic glucose metabolism in mice lacking Nur77. *Diabetes* 2009; 58: 2788-2796. (**Fig. 3B and D**)

[5]Jean Glrard. The inhibitory effects of insulin on hepatic glucose production are both direct and indirect. *Diabetes* 1 December 2006; 55 (Supplement_2): S65–S69.

[6]AD Cherrington. Banting lecture 1997. Control of glucose uptake and release by the liver in vivo. *Diabetes* 1999;48(6):1198-1214.

2. The role of GLP1 as a mediator of the actions of nesfatin-1 is very interesting and the MC4R-dependent action nicely fits with a previous study concerning the role of this receptor in the secretion of GLP1. However, there are other studies showing that direct stimulation of small intestinal Melanocortin-4-Receptors in mice and rats does not affect GLP-1 secretion (PMID: 34421821). Thus, I think it is important to strength this part of the manuscript:

a) Since the authors have found that nesfatin increases GLP1 while SHU9119 reduces GLP1, I would recommend measuring GLP1 in Figs 4e and 4f, which only focused on cAMP.

Response: We appreciate and agree with your suggestion. As requested, we have added content on GLP-1 to these figures (see **Fig. 4E and 4F, and 1st paragraph, page 10**).

b) I would recommend overexpressing MC4R and corroborate that GLP1 is augmented.

Response: We appreciate and agree with your comment. As requested, this experiment has been supplemented (see **Fig. S6F-J, 3rd paragraphs, page 10, 1st paragraphs, page 27**).

c) *The functional interaction between MC4R and NUCB2 has been established by using the pharmacological inhibitor SHU, which inhibits both MC3R and MC4R. I would recommend using a more specific approach inhibiting specifically MC4R (i.e. virogenetic tools, CRISPR) in the duodenum.*

Response: We agree with this suggestion. As requested, we performed scAAV inhibition of MC4R expression *in vivo* to further confirm that nesfatin-1 promotes GLP-1 secretion through MC4R mediation (see Fig. S6A-E, 3rd paragraphs, page 10 and 3rd paragraphs, page 26).

d) *Likewise, in vitro experiments should be done using more specific tools for MC4R and see whether the action of nesfatin on GLP1 levels is still reduced.*

Response: As requested, we used adenoviral vectors to inhibit and overexpress MC4R and observed the effect of nesfatin-1 on GLP-1 *in vitro* (see Fig. S6F-J, 3rd paragraphs, page 10; 1st paragraphs, page 27).

3. *Fig 7i is not convincing. The photo clearly shows that protein levels of MC4R are increased in mutant Nesfatin-1-treated mice compared to saline, but the figure represents identical values in these 2 groups. Also, these results should be corroborated in vitro by measuring MC4R in intestinal fragments and STC1 cells incubated with WT and mutant nesfatin-1.*

Response: We apologize for selecting this unsatisfactory band, which has been replaced by a repeated experimental band (see Fig. 7I). In addition, as requested, we conducted *in vitro* experiments by incubating intestinal segments and STC-1 cells with the WT and mutant nesfatin-1 proteins and observed the expression of MC4R (see Fig. 7J and K, and 1st paragraph, page 15).

4. *Since the study is focused in the effect of nesfatin-MC4R-GLP1 in intestinal cells, perhaps it would be relevant to also asses intestinal gluconeogenesis, which is activated by nutrients capable of fuelling systemic gluconeogenesis.*

Response: We understand your concerns and have added experiments. However, treatment

with nesfatin-1 did not result in altered gluconeogenesis in intestinal tissue (see **Rebuttal Fig. 1**).

Figure for reviewers removed

5. *The last part addressing the role of duodenal nesfatin-1 on HGP via the gut-brain-liver neural circuit is also relevant. However, in this section MC4R and GLP1 are totally missed. It would be important to measure both after the different treatments/surgery.*

Response: We fully understand your concerns. For this purpose, we measured the levels of MC4R and GLP1 after different treatments/surgery. However, the expression of intestinal MC4R and GLP1 was not affected by these therapeutic/surgery interventions (see **Rebuttal Fig. 2**). Therefore, this may be because MC4R and GLP1 are located upstream of the

gut-brain-liver neural circuit, as previously reported [1].

References

1. Yang M, et al. Duodenal GLP-1 signaling regulates hepatic glucose production through a PKC- δ -dependent neurocircuitry. *Cell Death Dis.* 2017; 8(2): e2609. doi: 10.1038/cddis.2017.28.

Figure for reviewers removed

Reviewer #2

These studies by Geng and colleagues propose a role for intestinal nesfatin-1 as a gut nutrient sensor that regulates hepatic glucose production (HGP) via a novel gut-brain-liver axis. Using duodenal infusions, pancreatic clamps, and a novel nesfatin-1 knockout rat line, the authors build a case for intestinal nutrient entry increasing gut nesfatin-1 levels, and this subsequently reduces HGP in a glucagon-like peptide-1 (GLP-1) dependent manner. Additional experiments suggest that nesfatin-1 mediates this effect by interacting with the melanocortin-4 receptor (MC4R). Furthermore, evidence is presented suggesting that gut nesfatin-1 action regulates HGP by engaging the central nervous system. Taken together, these studies propose a unique model for a gut hormone to act as a nutrient sensor via a novel inter-organ mechanism. Although these studies are novel, there are significant issues that should be addressed to more precisely interpret the results:

Major comments:

1. These studies rely extensively on a newly created nesfatin-1/NUCB2 knockout rat model. Additional details and data validating this model are needed. The only data shown for the generation of this model (Figure S6) only shows genomic DNA PCR results, and it is not quite clear what these results actually show. For example, PCR results from WT rats show a band at ~1.5kb and another at ~750kb. Are these bands expected, and why would the band at ~750kb that is present in WT rats also be present in the KO rats? Additional information along with showing that nesfatin-1 protein levels are reduced in various tissues in the KO rats is required.

Response: We apologize for this confusion. In Figure S8 (original Figure S6), Figure S8a shows the NUCB2 knockout strategy, where the wild type (WT) contains exons 6 to 9, while the knockout type removes exons 7 and 8. Primer F and Primer R are primers used for PCR amplification, located near exons 6 and 9, respectively (**see Figure S8A**).

Fig. S8B and Rebuttal Fig. 3 show the PCR amplification results. In first PCR validation, if only 630 bp band was amplified as homozygous, both 1335 bp and both 630 bp bands were amplified as heterozygous, and only 1335 bp band was amplified as WT (indicated by *). Because the larger 1335 bp bands are not readily amplified successfully, to ensure the correct

genotype, we designed a short-fragment primer, Primer-Wt/He-F, for identification of WT in exon 7, which produced a 672 bp band (indicated by #). Genotypes were identified by combining two pairs of primers: only 630 bp bands were homozygous; both 1335 bp and 672 bp were WT. Because 630 bp and 672 bp are very close, they are labeled in **Figure S8B and Rebuttal Fig. 3**.

In addition, as requested, we measured the protein expression of nesfatin-1 in various tissues of NUCB2 KO rats (see **Fig. S1 and 2nd paragraph, page 5**).

Figure for reviewers removed

2. *Figure 1b shows that a 30 min glucose infusion increases gut nesfatin-1 levels. This appears to be too short of an infusion for it to be a transcriptional event. The authors should discuss potential mechanisms for increased nesfatin-1 protein expression under such a relatively short time period. Furthermore, the authors should include information about the rate of duodenal glucose infusion (the methods only state that 25% glucose was infused, but the total mass of glucose infused should be provided). Also, the results in Figure 1c are presented as "Protein expression (%)", but it is not clear what this means (i.e., % of what?).*

Response: Thank you for your suggestion. As requested, we discussed the issue of increased NUCB2 expression during intestinal glucose infusion (see **2nd paragraph, page 17**). In addition, the rate and total amount of duodenal glucose infusion have been added to the methods section (see **3rd paragraph, page 24**). Finally, we have changed the expression of NUCB2 protein to relative value (see **Fig. 1C**).

3. For all clamp experiments, the entire time course of GIR should be presented. This is particularly important for assessing whether/to what degree the GIR was affected by the various duodenal infusions (glucose, lipid, nesfatin-1). As presented, only the last 20 minutes of GIR measurements during the 50 minute duodenal infusions are shown. The GIR throughout the entire 110 min clamp for all PEC experiments should be shown.

Response: As requested, the GIR *throughout* the entire 110 min of clamping has been added to the revised MS (see Fig. 1E and K, Fig. 2B and H, Fig. 3B, Fig. 5H and Fig. 7B).

4. Regarding the PEC clamp experiments, the authors should discuss why HGP was suppressed even though euglycemia was maintained and insulin levels did not increase from the basal to the clamp period. This is more relevant in the high fat diet experiments in which insulin levels actually dropped during the clamp experiments yet HGP was still somewhat suppressed in all groups.

Response: We fully understand your concerns. In this study, we employed PEC to suppress endogenous insulin by infusing somatostatin, thereby maintaining blood glucose and insulin levels close to baseline in each group of rats. On this basis, we used GIR to determine the differences in tissue insulin sensitivity among different groups of rats. In HFD-fed rats, due to the infusion of somatostatin inhibiting hyperinsulinemia, insulin levels during clamp steady state were low and close to their baseline levels. In addition, both somatostatin-induced glucagon inhibition and exogenous glucose infusion have inhibitory effects on HGP, as reported previously [1–3]. As requested, some of these contents have been included in the discussion (see 1st paragraph, page 18).

References

- 1) Abraham MA, Rasti M, Bauer PV, Lam TKT. Leptin enhances hypothalamic lactate dehydrogenase A (LDHA)-dependent glucose sensing to lower glucose production in high-fat-fed rats. *J Biol Chem.* 2018;293(11):4159-4166.
- 2) Cherrington AD. Banting Lecture 1997. Control of glucose uptake and release by the liver in vivo. *Diabetes.* 1999;48(5):1198-214.

3) Kowalchuk C, Teo C, Wilson V, Chintoh A, Lam L, Agarwal SM, Giacca A, Remington GJ, Hahn MK. In male rats, the ability of central insulin to suppress glucose production is impaired by olanzapine, whereas glucose uptake is left intact. *J Psychiatry Neurosci.* 2017;42(6):424-431.

5. Results from high fat diet fed rats require additional discussion because they suggest that the loss of nesfatin-1 does not just prevent the suppression of HGP by duodenal glucose but it actually impairs hepatic insulin resistance even further. This should be addressed in the Discussion.

Response: We appreciate your constructive comments. This content has been added to the discussion (see **1st paragraph, page 18**).

6. The duodenal nesfatin-1 infusion protocol requires additional information and validation. The authors state "To prevent an increase in circulating nesfatin-1 levels, a nesfatin-1 protein concentration of 100 ug/kg was selected for all subsequent experiments" (Page 7). One issue with this is that the 100 ug/kg is a concentration and not a rate of infusion. The rate of infusion should be provided. The other issue is that the authors should show whether this infusion does not increase circulating nesfatin-1 levels. This is an important result because it would validate the model that nesfatin-1 is acting locally in the gut.

Response: We appreciate and agree with your comment that the infusion rate of nesfatin-1 (2 µg/kg/min) has been added to the manuscript (see **1st paragraph, page 7 and 3rd paragraph, page 24**). In addition, we measured the circulating levels of nesfatin-1 and confirmed that intestinal nesfatin-1 did not affect the circulating levels of nesfatin-1 (see **Fig. S2H and 1st paragraph, page 7**).

7. There are several experiments that were conducted in NUCB2 KO rats but not in WT rats. This is a necessary control. For example, Fig. S2b shows that nesfatin-1 lowers glucose levels during a refeed in fast-refed NUCB2 KO rats. However, it is important to show: a) what are the glucose levels during the refeed period in a WT rat (are they comparable to those of the nesfatin-infused NUCB2 KO) and b) does nesfatin-1 further lower glucose levels during a

refeed in WT rats. Similar controls in WT rats should be done for the GTTs, ITTs.

Response: We appreciate your comment.

a) As requested, we performed fasting and refeeding experiments on WT rats (**Rebuttal Fig. 4a**) and found that intestinal nesfatin-1 infusion had a similar and slightly weaker effect on glucose levels in WT rats compared to NUCB KO rats (**Rebuttal Fig. 4b**), but had no effect on food intake (**Rebuttal Fig. 4c**).

b) As requested, we also performed GTT and ITT in WT rats and found that their results were similar to those of NUCB2 KO rats (**Rebuttal Fig. 4d and e**).

Given that nesfatin-1 is a secreted protein, to eliminate the interference of circulating nesfatin-1 on the results, we performed intestinal nesfatin-1 infusion into NUCB2-KO rats. We focused on the effect of intestinal nesfatin-1 on hepatic glucose metabolism under conditions of systemic nesfatin-1 deficiency. Because there are too many confounding factors in WT rats, we did not design a WT group. However, if you believe that including this group in the manuscript is essential, we are also willing to do so.

Figure for reviewers removed

8. Panaro *et al.* *Cell Metab* 2014 (reference 22) show that the MC4R is expressed in the basolateral side in gut enteroendocrine cells. This would make it difficult for a duodenal infusion of nesfatin-1 to interact with gut MC4R unless there is a mechanism for duodenally-infused nesfatin-1 to be transported from the luminal side to the basolateral side intact. Furthermore, the same study showed that activation of the intestinal MC4R does not increase GLP-1 levels in rats and only does so in mice. These discrepancies should be discussed. Panaro *et al.* *Cell Metab* 2014 (reference 22).

Response: We appreciate your constructive comments and have included relevant content in the discussion (see **1st paragraph, page 21**). In addition, to further validate the relationship between MC4R and GLP-1, we performed additional experiments. The results showed that the activation of MC4R by nesfatin-1 can indeed affect the level of GLP-1 in rats (see **Fig. S6F and 3rd paragraphs, page 10**). Therefore, we believe that the difference between Panaro *et al.* and our results is mainly due to different experimental methods: 1) Panaro *et al.* used mesenteric artery infusion for drug administration, while we used local stimulation in the intestine; 2) Panaro *et al.* used short-term infusion (8 minutes), while we stimulated for 24 h; 3) Panaro *et al.* measured circulating GLP-1 levels, while we measured GLP-1 concentrations in local tissue cell mediators; and 4) the methods for measuring GLP-1 are different. However, further *in vitro* and *in vivo* research is needed. These contents have been added to the discussion (see **1st paragraph, page 19**).

9. Fig. 5 shows increased nesfatin-1 and MC4R expression in duodenal epithelium following duodenal nesfatin-1 infusion. The assumption is that the increased nesfatin-1 is from the exogenous infusion since these experiments were conducted in NUCB2 KO rats. The authors should clarify this (this also partially addresses point #7 above). Furthermore, the authors should also provide data obtained from WT rats. This will show whether NUCB2 KO rats have lower baseline gut MC4R expression.

Response: We appreciate and understand your feedback. Considering that nesfatin-1 is a secreted protein, to eliminate the interference of circulating nesfatin-1 on the results, we performed intestinal nesfatin-1 infusion in Nucb2 KO rats. We focused on the effect of intestinal nesfatin-1 on hepatic glucose metabolism under conditions of systemic nesfatin-1 deficiency. Given that there are too many confounding factors in WT rats, we did not design a WT group. In addition, as requested, we observed basal MC4R expression in the guts of WT and NUCB2 KO rats and found that gut MC4R expression in NUCB2 KO rats was significantly lower than that in WT rats (**see Rebuttal Fig. 5**).

Figure for reviewers removed

10. *The experimental interventions used to test the gut-brain-liver connection should include validations. Evidence that tetracaine inhibited neurotransmission and MK-801 inhibited NMDA signaling should be provided. Also, the authors should specify whether these experiments were conducted in WT or NUCB2KO rats.*

Response: We appreciate your comment and understand your concern about this issue. As requested, references regarding the inhibition of vagus nerve transmission by tetracaine and the inhibition of NMDA signaling by MK-801 have been added to the revised MS [1-5] (**see 2nd paragraph, page 15 and 1st paragraph, page 16**). Furthermore, we apologize for any confusion caused by the experimental model. In fact, these experiments were performed on NUCB2 KO rats (**see Fig. S7B, 2nd paragraph, page 15 and 1st paragraph, page 16**).

References

- 1) Kokorovic A, Cheung GW, Breen DM, Chari M, Lam CK, Lam TK. Duodenal mucosal protein kinase C-delta regulates glucose production in rats. *Gastroenterology* 2011;141:1720-1727.
- 2) Rasmussen BA, Breen DM, Luo P, Cheung GW, Yang CS, Sun B, Kokorovic A, et al. Duodenal activation of cAMP-dependent protein kinase induces vagal afferent firing and lowers glucose production in rats. *Gastroenterology* 2012;142:834-843 e833.
- 3) Wang PY, Caspi L, Lam CK, Chari M, Li X, Light PE, Gutierrez-Juarez R, et al. Upper intestinal lipids trigger a gut-brain-liver axis to regulate glucose production. *Nature* 2008;452:1012-1016.
- 4) Breen DM, Rasmussen BA, Kokorovic A, Wang R, Cheung GW, Lam TK. Jejunal nutrient sensing is required for duodenal-jejunal bypass surgery to rapidly lower glucose concentrations in uncontrolled diabetes. *Nat Med* 2012;18:950-955.
- 5) Yang M, Wang J, Wu S, Yuan L, Zhao X, Liu C, Xie J, et al. Duodenal GLP-1 signaling regulates hepatic glucose production through a PKC-delta-dependent neurocircuitry. *Cell*

11. *In general, all figure legends should describe the statistical test(s) used for the specific comparisons shown in that figure.*

Response: Thank you for your reminder. As requested, the statistical test(s) used for the specific comparisons are shown in the **figure legends**.

Minor comments:

1. *In Page 5, the authors state "Throughout the PEC period, blood glucose, insulin, free fatty acid (FFA), and triglyceride (TG) levels remained stable at basal levels (Fig. S1a-e)" when referring to results obtained from experiments using duodenal saline/glucose infusions. However, the legend for Fig. S1a-e indicates that these are values obtained from experiments in which duodenal nesfatin-1 was infused.*

Response: We apologize for this mistake and appreciate your reminder. This has been revised (see **Figure legend for Fig. S2**).

2. *In Page 5 the authors also state "In contrast, NUCB2 KO rats demonstrated a substantial reduction in GIR (Fig. 1e and f) and a corresponding increase in HGP (Fig. 1g and h)". It would be more accurate to state "a corresponding impairment in the suppression of HGP". A similar modification should be made in Page 6, in the sentence "Conversely, lipid-infused NUCB2 KO rats demonstrated a notable decrease in GIR (Fig. 1k and l) and a significant increase in HGP (Fig. 1 m-n)".*

Response: Thank you for your reminder. As requested, this sentence has been revised (see **2nd paragraph, page 5 and 2nd paragraph, page 6**).

3. *There appears to be an incorrect statement in Page 7, "We hypothesized that duodenal nesfatin-1 may interfere with nutrient-sensing-related mechanisms activated by refeeding, thereby further inhibiting HGP and reducing blood glucose." Wouldn't the hypothesis be*

that nesfatin-1 "contributes to" and not interferes with nutrient-sensing-related mechanisms?

Response: Thank you for your correction, and it has been revised (see 2nd paragraph, page 7).

Dear Dr Yang,

Thank you for submitting your revised manuscript (EMBOJ-2023-116057R) to The EMBO Journal, as well as for your patience with our response at this time of the year. Your amended study was sent back to the referees for their re-evaluation, and we have received re-comments from both of them, which I enclose below. As you will see, the experts stated that the work has been substantially improved by the revisions and they are now in favour of publication, pending minor revision.

Thus, we are pleased to inform you that your manuscript has been accepted in principle for publication in The EMBO Journal.

Please consider the remaining issues of referee #2 carefully and amend the manuscript accordingly by complementary statistical analysis, or alternatively revisiting the discussion of the results.

We also now need you to take care of a number of minor issues related to formatting and data presentation as detailed below, which should be addressed at re-submission.

Please contact me at any time if you have additional questions related to below points.

As you might have seen on our web page, every paper at the EMBO Journal now includes a 'Synopsis', displayed on the html and freely accessible to all readers. The synopsis includes a 'model' figure as well as 2-5 one-short-sentence bullet points that summarize the article. I would appreciate if you could provide this figure and the bullet points.

Thank you for giving us the chance to consider your manuscript for The EMBO Journal.
I look forward to your final revision.

Again, please contact me at any time if you need any help or have further questions.

Kind regards,

Daniel Klimmeck

>> Author Contributions: add information on author contributions. Note that CRediT has replaced the traditional author contributions section as of now because it offers a systematic machine-readable author contributions format that allows for more effective research assessment. and use the free text boxes beneath each contributing author's name to add specific details on the author's contribution.

More information is available in our guide to authors.
<https://www.embopress.org/page/journal/14602075/authorguide>

>> Introduce ORCID IDs for all corresponding authors (L. L.) via our online manuscript system. Please see below for additional information.

>> Rename the current 'Declaration of Interests' section to 'Disclosure and Competing Interests Statement'.

>> Reagents and Tools Table: Add a completed Reagents and Tools table to the Methods, listing key reagents, experimental models, software and relevant equipment. The table needs to be uploaded in the correct format using the template provided on our webpage.

>> Appendix file: the file with suppl. figs and tables should be renamed "Appendix". A table of contents with page numbers should be added to the first page, and the red font should be removed. The figures and tables should be corrected to "Appendix Figure S1" etc. and "Appendix Table S1" etc. .

>> Source data: files should be uploaded as one (zipped) file per figure.

>> Data availability section: please state 'no data amenable to large-scale repository deposition are included in this study.' Adjust the Author Checklist accordingly by unchecking the 'primary datasets' option in the Data availability section.

>> References: please adjust the reference format to EMBO Journal format, 10 authors et al, and place References after the Discussion, before figure legends.

>> Avoid textual redundancy with the earlier article Geng et al. 2023 (PMID:38156278), in the introduction, results and discussion sections.

>> Consider additional changes and comments from our production team as indicated below:

Figure legends

1. Please note that the exact p values are not provided in the legends of figures 1c, e-h, k-n; 2b-f, h-k; 3b-e, g-h; 4c-f; 5c-d, f-k, m-p; 6j; 7b-k.

2. Please indicate the statistical test used for data analysis in the legend of figure 4b.

Please

Please note that as of January 2016, our new EMBO Press policy asks for corresponding authors to link to their ORCID iDs. You can read about the change under "Authorship Guidelines" in the Guide to Authors here: <http://emboj.embopress.org/authorguide>

In order to link your ORCID iD to your account in our manuscript tracking system, please do the following:

1. Click the 'Modify Profile' link at the bottom of your homepage in our system.
2. On the next page you will see a box half-way down the page titled ORCID*. Below this box is red text reading 'To Register/Link to ORCID, click here'. Please follow that link: you will be taken to ORCID where you can log in to your account (or create an account if you don't have one)
3. You will then be asked to authorise Wiley to access your ORCID information. Once you have approved the linking, you will be brought back to our manuscript system.

We regret that we cannot do this linking on your behalf for security reasons. We also cannot add your ORCID iD number manually to our system because there is no way for us to authenticate this iD number with ORCID.

Thank you very much in advance.

Referee #1:

The authors have addressed all my concerns and I have no further comments.

Referee #2:

This revised manuscript addresses many of the comments raised in the original review. However, there is still one issue to address.

The statistical analyses for some of the results are not appropriate. Experiments in which there are two treatments and two genotypes (Figures 1E-O, 2D-L) describe using "one-way ANOVA followed by multiple t-tests with Bonferroni Correction". First, it is not clear how "multiple t-tests" can be done following a one-way ANOVA. Second, since there are two independent variables (treatment and genotype), a one-way ANOVA is not appropriate (a two-way ANOVA is more appropriate). The same also holds for Figure S3B where time and treatment are independent variables. For the time course GIR, time is also an independent variable, so a three-way ANOVA would be appropriate. This is important given the relatively small number of replicates (n=6). The authors should run the appropriate tests and indicate whether there are/are not interactions between the variables.

For other appropriate statistical analyses in which the authors state that a "one-way ANOVA followed by multiple t-tests with Bonferroni Correction" was used, the authors should clarify what they mean by this since a t-test and a one-way ANOVA are two separate and different kinds of tests.

Minor: AUC for an ITT is not informative and can be omitted (Figure S3E).

The authors addressed the remaining editorial issues.

Dear Dr Yang,

Thank you for submitting the revised version of your manuscript. I have now evaluated your amended manuscript and concluded that the remaining minor concerns have been sufficiently addressed.

I am thus pleased to inform you that your manuscript has been accepted for publication in the EMBO Journal.

Related I would like to hereby ask your consent on keeping the referee figures included in this file.

On a different note, I would like to alert you that EMBO Press offers a format for a video-synopsis of work published with us, which essentially is a short, author-generated film explaining the core findings in hand drawings, and, as we believe, can be very useful to increase visibility of the work. Please see the following link for representative examples and their integration into the article web page:

<https://www.embopress.org/doi/full/10.15252/emj.2019103932>

Best regards,

Daniel Klimmeck

Daniel Klimmeck, PhD
Senior Editor
The EMBO Journal
EMBO
Postfach 1022-40
Meyerhofstrasse 1
D-69117 Heidelberg
contact@embojournal.org
Submit at: <http://emboj.msubmit.net>